# Lsr2 acts as a cyclic di-GMP receptor that promotes keto-mycolic acid synthesis and biofilm formation in mycobacteria

Xiaocui Ling [1], Xiao Liu[1], Kun Wang[1], Minhao Guo[1], Yanzhe Ou[1], Danting Li[1], Yulin Xiang[1], Jiachen Zheng[1], Lihua Hu[1], Hongyun Zhang[1] & Weihui Li [1] ✉

Cyclic di-GMP (c-di-GMP) is a second messenger that promotes biofilm formation in several bacterial species, but the mechanisms are often unclear. Here, we report that c-di-GMP promotes biofilm formation in mycobacteria in a manner dependent on the nucleoid-associated protein Lsr2. We show that c-di-GMP specifically binds to Lsr2 at a ratio of 1:1. Lsr2 upregulates the expression of HadD, a (3R)-hydroxyacyl-ACP dehydratase, thus promoting the synthesis of keto-mycolic acid and biofilm formation. Thus, Lsr2 acts as a c-di-GMP receptor that links the second messenger's function to lipid synthesis and biofilm formation in mycobacteria.

Cyclic di-GMP (c-di-GMP), a conserved second messenger in bacteria, regulates several important physiological processes, including biofilm formation[1], motility[2], and virulence[3]. Generally, under high levels of intracellular c-di-GMP, bacteria tend to grow in biofilms rather than as a planktonic cells[4]. Recently, it has been reported that c-di-GMP promotes biofilm formation mediated by elongation factor P in *Acinetobacter baumannii*[5]. c-di-GMP also interacts with its effector BpfD to regulate biofilm formation in *Shewanella putrefaciens*[6]. However, the regulatory mechanism by which c-di-GMP regulates biofilm formation and the receptor involved in mycobacteria remain unknown.

Biofilms are a critical structured community for bacterial adaptation to harsh environmental conditions. Several mycobacterial species, including *Mycobacterium tuberculosis*, have been shown to form biofilms in vitro[7,8]. Mycobacteria have a lipid bilayer cell wall structure in which lipid components account for 60% (wt/wt) of the cell wall weight[9]. A variety of lipids have been reported to play important roles in mycobacterial biofilm formation[10–12]. The extracellular matrix of mycobacterial biofilms is rich in mycolic acid, an important lipid component of the cell wall[13,14]. Moreover, an increased c-di-GMP concentration has been shown to increase the production of keto-mycolic acid (keto-MA) in *Mycobacterium bovis* BCG[15]. However, the signal regulatory pathway by which c-di-GMP regulates keto-MA synthesis is still unclear.

Lsr2 is a nucleoid-associated protein (NAP) responsible for maintaining a highly organized chromosome structure and transcriptional regulation in mycobacteria, including *M. smegmatis*[16] and *M. tuberculosis*[17,18]. As a global regulator, Lsr2 regulates biofilm formation[19–22] and the cell cycle[23] in *M. smegmatis*. In *M. tuberculosis*, Lsr2 extensively regulates antibiotic resistance[24], oxidative stress[25], and virulence[18]. Mutation of Lsr2 led to smooth colony generation and deficient biofilm formation in *M. smegmatis*[19,21]. However, the regulatory pathway by which Lsr2 regulates biofilm formation remains largely obscure, especially in terms of upstream regulatory signals and downstream target genes.

HadD is a (3R)-hydroxyacyl-ACP dehydratase of the fatty acid synthase type II (FAS-II) system. It has been reported that HadD is involved in FAS-II elongation cycles during the synthesis of MA in mycobacteria[26,27]. *hadD* deletion alters the composition of MA and its relative lipids, leading to impaired rough colony generation and biofilm formation[26]. However, the upstream regulatory pathway of HadD in mycobacteria is still unclear.

Here, we found that Lsr2 is a receptor for c-di-GMP, with a binding ratio of 1:1. Lsr2 plays a crucial role in c-di-GMP by regulating colony morphology and biofilm formation in mycobacteria. In addition, Lsr2 positively regulated biofilm formation by triggering the expression of HadD to synthesize keto-MA, a lipid of the mycobacterial cell wall. Moreover, our study demonstrates that c-di-GMP stimulates Lsr2 activity to positively regulate the expression of *hadD*. Therefore, our study revealed the signaling pathway through which the c-di-GMP combines with the receptor Lsr2 to control biofilm formation. This

[1]State Key Laboratory for Conservation and Utilization of Subtropical Agro-bioresources, College of Life Science and Technology, Guangxi University, Nanning 530004, China. ✉e-mail: lwhlbx@163.com

finding reveals the links among second messenger, lipids, and mycobacterial biofilm formation.

## Results

### High levels of c-di-GMP affect colony morphology and biofilm formation of *M. smegmatis*

In our preliminary experiments, we found that overexpression of the diguanylate cyclase (DGC) gene *ydeH* from *Escherichia coli* in *M. smegmatis* resulted in bacterial adhesion to pipette tips. To explore the molecular mechanism underlying this strange phenomenon, we constructed a high c-di-GMP-content *M. smegmatis* strain by overexpressing *ydeH* (*ydeH*) and a control strain (*ydeH*(mut)) by overexpressing the mutant *ydeH* gene[28]. Surprisingly, high levels of c-di-GMP significantly altered the bacterial phenotypes, including increasing adhesion (Fig. 1A), sedimentation (Fig. 1B), and pellicle at the air–liquid surface (Fig. 1C). Furthermore, under a light microscope, the accumulation of *ydeH* resulted in robust crystal violet staining (Fig. 1D). A similar result was found by scanning electron microscopy, as *ydeH* bacteria adhered together (Fig. 1E). In addition, the overexpression of *ydeH* changed the colony phenotype on LB or 7H10 agar plates. The *ydeH* strain formed a typical wrinkled colony that was rougher than the smooth and wet colonies of the *ydeH*(mut) strain (Fig. 1F, G).

Changes in the surface morphology of bacterial colonies are often accompanied by changes in biofilm formation. We further investigated the effect of c-di-GMP on biofilm formation in *M. smegmatis*. As shown in Fig. 1H, the *ydeH* strain formed a strong biofilm and denser structured pellicle at the air–liquid surface when cultured in modified M63 medium. In contrast, the *ydeH*(mut) and WT strains exhibited sparse structured pellicles. The results were consistent with the biofilm biomass assay with crystal violet staining (Fig. 1I). And there was no significant difference in the growth of WT, *ydeH,* and *ydeH*(mut) strains, indicating that the biofilm formation was not related to the growth (Fig. S1A). Biofilm formation is associated with the motility of bacteria[21,29], so the motility of the WT, *ydeH* and *ydeH*(mut) strains was further investigated on semisolid M63 medium. The motility of the *ydeH* strain was reduced compared to that of the WT and *ydeH*(mut) strains (Fig. S2).

In conclusion, our results indicate that high levels of c-di-GMP can increase adhesion and sedimentation, alter wrinkled colonies, promote biofilm formation, and inhibit the motility of *M. smegmatis*.

### c-di-GMP affects biofilm formation in a manner dependent on Lsr2$_{Msm}$ in *M. smegmatis*

To investigate the regulatory mechanism of c-di-GMP on biofilm formation and identify the effector, we generated a transposon insertion mutant library of *M. smegmatis* with high levels of c-di-GMP. DNA sequencing and analysis of a colony-altered mutant revealed that a transposon was inserted into the *lsr2$_{Msm}$* gene (MSMEG_6092). Therefore, we speculate that c-di-GMP may regulate biofilm formation through Lsr2$_{Msm}$, a nucleoid-associated protein (NAP). To confirm our hypothesis, we constructed an *lsr2$_{Msm}$* deletion mutant strain (verified by RT-qPCR, Fig. S3) and several recombinant strains by complementing the *lsr2$_{Msm}$* gene or overexpressing the *ydeH* gene in the *lsr2$_{Msm}$*KO strain (*lsr2$_{Msm}$*Com and *lsr2$_{Msm}$*KO/*ydeH*), with the *lsr2$_{Msm}$* deletion mutant strain had an empty pMV261 vector as a control (*lsr2$_{Msm}$*KO). As shown in Fig. S4A–C, the colony phenotype of the *lsr2$_{Msm}$*com strain was partially restored to that of the WT strain. There was no significant difference in biofilm formation or quantitative biofilm biomass between the *lsr2$_{Msm}$*com and WT strains. The rough colony *ydeH*-overexpressing strains formed stronger biofilms than the smooth colony *lsr2$_{Msm}$*KO strain, which formed a very weak biofilm (Fig. 2A–C). The *lsr2$_{Msm}$*KO/*ydeH* strain formed a smooth colony and poor biofilm similar to those of the *lsr2$_{Msm}$*KO strain (Fig. 2A–C). This result was consistent with the quantitative biofilm biomass results of

crystal violet staining (Fig. 2D). And there was no significant difference in the growth of these strains, indicating that the biofilm formation was not related to the growth (Fig. S1B, C). These indicate that high levels of c-di-GMP can significantly alter the colony surface morphology of the WT strain of *M. smegmatis* but have no effect on the *lsr2$_{Msm}$*KO strain. In addition, we constructed the recombinant strains with downregulated gene expression using the CRISPRi (verified by RT-qPCR, Fig. S5) to examine the effect of other receptors of c-di-GMP including DevR$_{Msm}$ and LtmA$_{Msm}$ on colony morphology and biofilm formation in *M. smegmatis*. The results show that there were no significant difference in growth, colony morphology and biofilm formation among the pLJR962, *devR$_{Msm}$* CRISPRi and *ltmA$_{Msm}$* CRISPRi strains (Figs. S1D and S6).

Therefore, our data suggest that c-di-GMP largely relies on the NAP Lsr2$_{Msm}$ to regulate colony phenotype and biofilm formation in *M. smegmatis*.

### Lsr2$_{Msm}$ is a c-di-GMP receptor

As a signaling molecule, c-di-GMP exerts its function through its downstream receptor, while it regulates biofilm formation in a manner dependent on Lsr2$_{Msm}$. Therefore, Lsr2$_{Msm}$ might be a potential receptor for c-di-GMP, and we further validated this possibility through an isothermal titration calorimetry (ITC) assay. As shown in Fig. 2E, the raw data for titration of c-di-GMP against Lsr2$_{Msm}$ indicating that the interaction is exothermic (upper plots). The integrated heat measurements were shown in the lower plots. The binding stoichiometry between c-di-GMP and Lsr2$_{Msm}$ was 1:1 ($n = 0.869 \pm 0.03$), and the binding affinity of the interaction (Kd) was $9.644 \times 10^{-8} \pm 0.17$ M.

This result demonstrates that Lsr2$_{Msm}$ can directly interact with c-di-GMP and that Lsr2$_{Msm}$ is a receptor for c-di-GMP to control biofilm formation.

### *hadD$_{Msm}$* positively regulates biofilm formation in *M. smegmatis*

To further explore the regulatory pathway by which c-di-GMP and Lsr2$_{Msm}$ control biofilm formation, we constructed a whole-genome CRISPRi library using the dCas9-sgRNA complex to target and repress gene expression in high c-di-GMP-content *M. smegmatis*[30,31]. A recombinant strain (*hadD$_{Msm}$* CRISPRi) with a significantly altered colony morphology was screened on 200 ng/mL anhydrotetracycline (ATc) plates (Fig. S7), and further sequencing revealed that the gene target with CRISPRi was *hadD$_{Msm}$* (MSMEG_0948). HadD$_{Msm}$ is a 3R-hydroxyacyl-ACP dehydratase of the type II fatty acid synthase group that catalyzes the elongation step during MA biosynthesis[26].

The colony phenotype of the *hadD$_{Msm}$* deletion strain (*hadD$_{Msm}$*KO) and complementary strain (*hadD$_{Msm}$*Com) further confirmed that the altered colony phenotype was caused by the deletion of *hadD$_{Msm}$* (Fig. S8A). Furthermore, as shown in Fig. 3A, the *hadD$_{Msm}$*KO strain exhibited a smooth colony similar to that of the *lsr2$_{Msm}$*KO strain, and the *hadD$_{Msm}$* complementary strain morphology was restored to the WT phenotype. In addition, biofilm formation observation and quantitation for these three strains showed that the *hadD$_{Msm}$* deletion mutant formed a fragile biofilm after a delay, and the WT and *hadD$_{Msm}$*Com strains exhibited significant wrinkled pellicle biofilm formation (Fig. S8B, C). And there was no significant difference in the growth of these strains, indicating that the biofilm formation was not related to the growth (Fig. S1E).

These results indicate that *hadD$_{Msm}$* is important for biofilm formation in *M. smegmatis*.

### Lsr2$_{Msm}$ positively regulates biofilm formation by directly regulating *hadD$_{Msm}$*

Lsr2$_{Msm}$ and *hadD$_{Msm}$* perform similar functions in regulating biofilm formation and Lsr2$_{Msm}$ is a NAP with extensive regulatory functions. Therefore, we hypothesized that Lsr2$_{Msm}$ affects biofilm formation through its regulatory effect on *hadD$_{Msm}$*. To verify our hypothesis, an

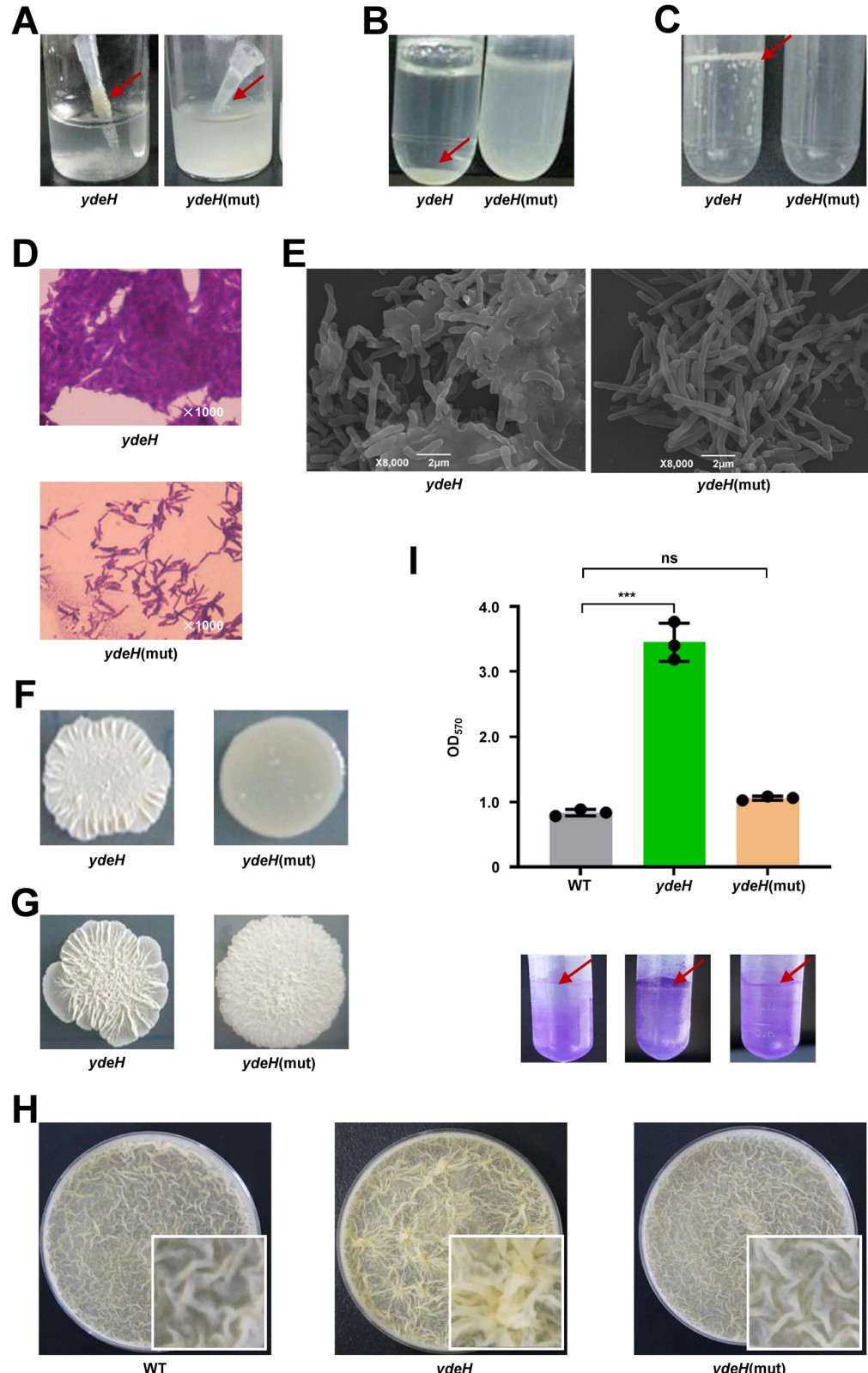

**Fig. 1 | c-di-GMP controls biofilm formation by *M. smegmatis*. A** Adhesion of the *ydeH*, and *ydeH*(mut) strains. **B** Sedimentation rate of the *ydeH*(mut), and *ydeH* strains. **C** Pellice formation of the *ydeH*, and *ydeH*(mut) strains. **D** The cells morphology of the *ydeH*, and *ydeH*(mut) strains after staining with 2% crystal violet under light microscope. **E** Scanning electron microscopy assays of cell morphology for *ydeH*, and *ydeH*(mut) strains. The images were taken at 8000x magnification. The experiments of (**D**–**E**) were performed three times and the representative images were shown. **F** Colonial morphology of the *ydeH* and *ydeH*(mut) strains on LB agar plates. **G** Colonial morphology of the *ydeH* and *ydeH*(mut) strains on 7H10 agar plates. **H** Biofilm formation at the air–liquid surface of the wild type, *ydeH* and *ydeH*(mut) strains. **I** Quantitation of biofilm biomass by crystal violet staining of the wild type, *ydeH* and *ydeH*(mut) strains (*n* = 3, biological replicates). Two-tailed t-tests were performed for statistical analysis (***$p$ = 0.0001). Data were presented as mean ± SD. The source data were provided in the Source data file.

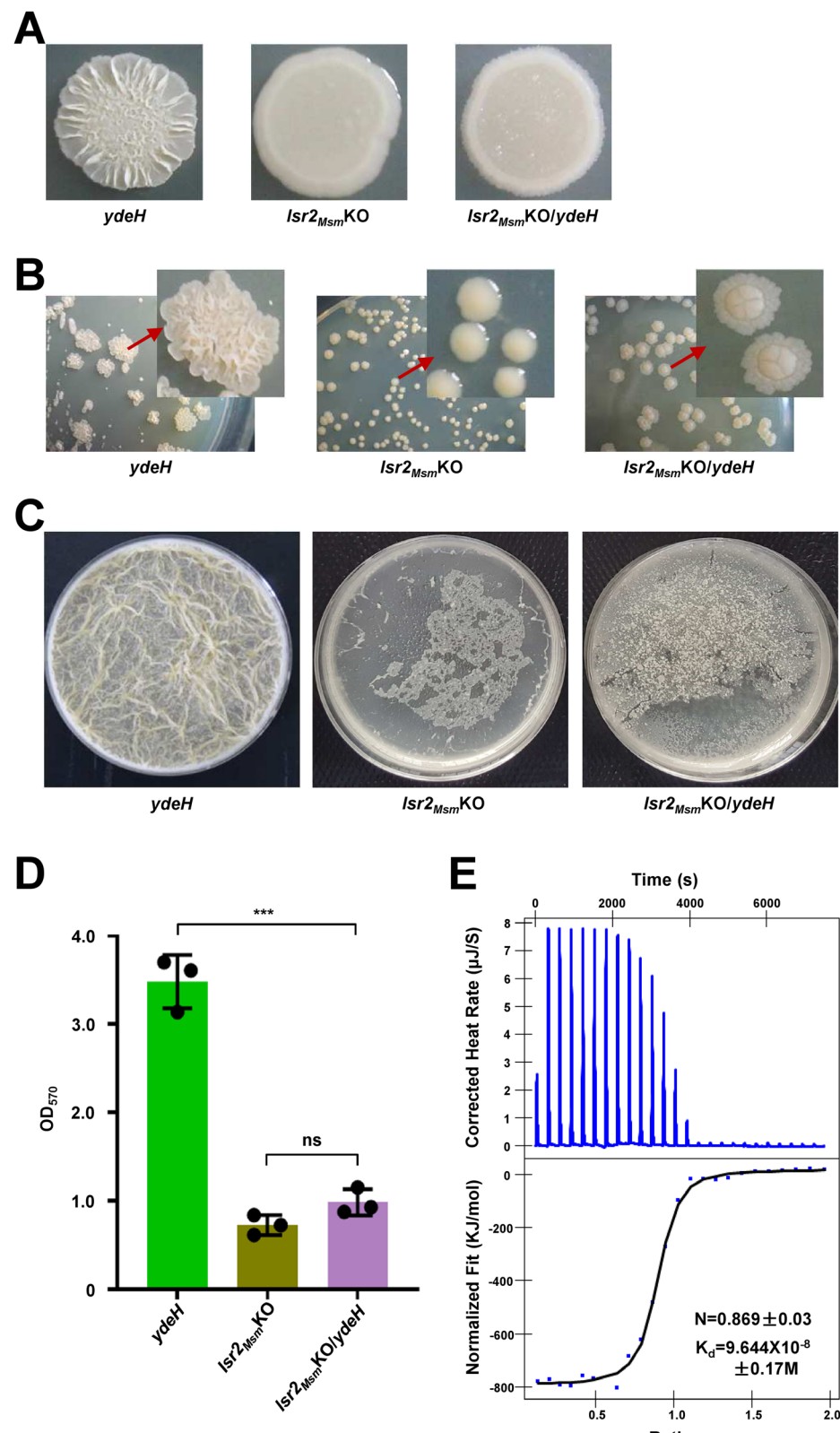

**Fig. 2 | c-di-GMP control biofilm formation via Lsr2$_{Msm}$, and Lsr2$_{Msm}$ is a c-di-GMP receptor. A** Spot colony morphology of the wide type, *lsr2$_{Msm}$* knock-out and *lsr2$_{Msm}$*KO/*ydeH* strains. **B** Single colony morphology of the *ydeH*, *lsr2$_{Msm}$*KO and *lsr2$_{Msm}$*KO/*ydeH* strains on 7H10 medium. **C** Biofilm formation at the air–liquid surface of the *ydeH*, *lsr2$_{Msm}$*KO and *lsr2$_{Msm}$*KO/*ydeH* strains. **D** Biofilm biomass quantitation of the *ydeH*, *lsr2$_{Msm}$*KO and *lsr2$_{Msm}$*KO/*ydeH* strains by crystal violet staining (*n* = 3, biological replicates). Two-tailed t-tests were performed for statistical analysis (***$p$ = 0.0002). Data were presented as mean ± SD. **E** ITC assays for the interaction between Lsr2$_{Msm}$ and c-di-GMP. Original titration data and integrated heat measurements were shown in the upper and lower plots. The source data were provided in the Source data file.

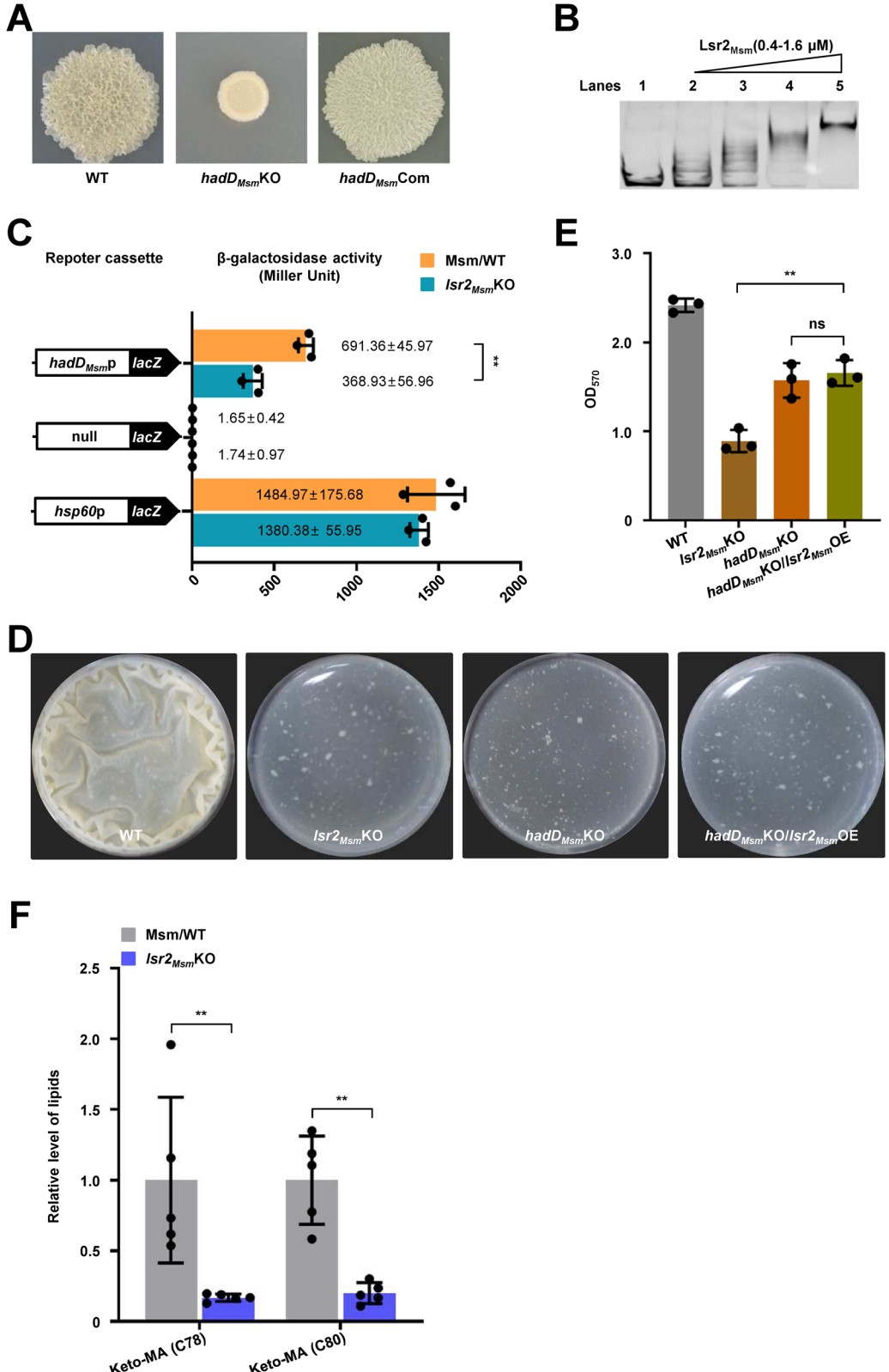

EMSA conducted in vitro confirmed that Lsr2$_{Msm}$ could directly regulate $hadD_{Msm}$ (Figs. 3B and S9). In contrast, Lsr2$_{Msm}$ mutant protein loses the DNA-binding activity (Fig. S10A, B). A β-galactosidase activity assay was conducted to further characterize the regulatory effect of Lsr2$_{Msm}$ on the expression of $hadD_{Msm}$. The expression level of $hadD_{Msm}$p-$lacZ$ in the WT strain was significantly higher than that in the $lsr2_{Msm}$KO strain (Fig. 3C). However, there was no significant difference between the WT and $lsr2_{Msm}$KO strains in the expression of $lacZ$ when the irrelevant control, $ms5038$p and $ms4718$p, were used as promoters (Fig. S11). These results indicate that Lsr2$_{Msm}$ directly positively regulates the transcription of $hadD_{Msm}$.

Next, we further identified whether Lsr2$_{Msm}$ controls biofilm formation by regulating $hadD_{Msm}$. $lsr2_{Msm}$ was overexpressed in the $hadD_{Msm}$ deletion mutant ($hadD_{Msm}$KO/$lsr2_{Msm}$OE) for colony

Fig. 3 | Lsr2$_{Msm}$ regulates biofilm formation in a manner dependent on *hadD$_{Msm}$*. **A** Spot colony morphology of the WT, *hadD*$_{Msm}$KO and its *hadD*$_{Msm}$ complementary strains. **B** EMSA assays for the *hadD*$_{Msm}$ promoter DNA-binding activity of Lsr2$_{Msm}$. *hadD*$_{Msm}$p was co-incubated with increasing amounts of Lsr2$_{Msm}$ (lanes 2–5). The experiment was performed three times and the representative image was shown. **C** The β-galactosidase activity experiment to assay for the effect of the Lsr2$_{Msm}$ on the expression of *hadD* by overexpressing *hadD*$_{Msm}$p-*lacZ* plasmid in the wild-type *M. smegmatis* strain (Msm/WT) and *lsr2*$_{Msm}$ knock-out strain (*lsr2*$_{Msm}$KO) (*n* = 3, biological replicates). None promoter-*lacZ* and *hsp60*p-*lacZ* were used as controls. The data were presented as Miller units on the right panel. Left column: schematic representation of each plasmid used to construct recombinant strains. Two-tailed Student's t-tests were performed for statistical analysis of three independent biological experiments (\**p* = 0.0011). **D** Biofilm formation at the air–liquid surface of the WT, *lsr2*$_{Msm}$KO, *hadD*$_{Msm}$KO and its *lsr2*$_{Msm}$ overexpression strains. **E** Quantitation of biofilm biomass by crystal violet staining (*n* = 3, biological replicates). Two-tailed t-tests were performed for statistical analysis (\*\**p* = 0.0023). **F** The quantity of keto-MA between the WT and *lsr2*$_{Msm}$KO *M. smegmatis* strains detected by lipidomic assays (*n* = 6, biological replicates). The significant differences were determined by unpaired two-tailed Student's t-test (\*\**p* = 0.0024, \*\**p* = 0.0033). Data were presented as mean ± SD of (**C**), (**D**), (**F**). The source data were provided in the Source data file.

morphology and biofilm detection. Notably, the *hadD*$_{Msm}$KO/*lsr2*$_{Msm}$OE strain exhibited a smooth colony phenotype similar to that of the *hadD*$_{Msm}$KO and *lsr2*$_{Msm}$KO strains (Fig. S12A, B). Furthermore, the three strains formed fragile biofilms at the air–liquid interface, and their biofilm biomass was significantly lower than that of the WT strain (Fig. 3D, E). And there was no significant difference in the growth of these strains, indicating that the biofilm formation was not related to the growth (Fig. S1F). Lsr2$_{Msm}$ positively regulated biofilm formation, but the overexpression of *lsr2*$_{Msm}$ had no effect on biofilm formation of the *hadD*$_{Msm}$KO/*lsr2*$_{Msm}$OE strain due to the deletion of *hadD*$_{Msm}$.

In conclusion, these results indicate that Lsr2$_{Msm}$ regulates biofilm formation by directly upregulating the expression of *hadD*$_{Msm}$.

## Lsr2$_{Msm}$ regulates keto-MA synthesis to control biofilm formation in *M. smegmatis*

Lsr2$_{Msm}$ is involved in biofilm formation. To analyze the role of Lsr2$_{Msm}$ in biofilm formation in depth, lipidomic analysis was performed on the *lsr2*$_{Msm}$KO strain. As shown in Fig. S13, significant changes were detected in the levels of 484 molecules, among which 241 molecules were significantly enriched in the *lsr2*$_{Msm}$KO strain (fold change >1.2, *p* < 0.05), and 243 molecules were significantly depleted (fold change <0.83, *p* < 0.05) compared with the WT strain. Further analysis showed that in the *lsr2*$_{Msm}$ deletion mutant, keto-MA (C78, C80) were significantly depleted (Fig. 3F). This result was consistent with the decrease in the context of long-chain keto-MA in the *hadD*-deficient strain, which resulted in poor biofilm formation, as previously reported[27].

In general, our data suggest that Lsr2$_{Msm}$ regulates the synthesis of keto-MA by *hadD*$_{Msm}$ to control biofilm formation.

## c-di-GMP relies on *hadD*$_{Msm}$ to regulate biofilm formation in *M. smegmatis*

*hadD*$_{Msm}$ is a target gene directly regulated by Lsr2$_{Msm}$, and Lsr2$_{Msm}$ is a c-di-GMP receptor that regulates biofilm formation. Next, we investigated whether the regulation of biofilm formation by c-di-GMP depends on the *hadD*$_{Msm}$ gene. An *ydeH* overexpression strain was constructed bearing the *hadD*$_{Msm}$ deletion mutation (*hadD*$_{Msm}$KO/*ydeH*). The colony phenotype results showed that *hadD*$_{Msm}$KO/*ydeH* strain formed a smoother colony than the WT strain and a rougher colony than the *hadD*$_{Msm}$KO strain. In contrast, the *ydeH* strain formed the roughest colony (Figs. 4A and S14). Biofilm formation and quantitation assays showed that the *ydeH* strain formed robust biofilms, but the *hadD*$_{Msm}$KO and *hadD*$_{Msm}$KO/*ydeH* strains formed weak biofilms (Fig. 4B, C). And there was no significant difference in the growth of these strains, indicating that the biofilm formation was not related to the growth (Fig. S1G). The expression of *ydeH* did not restore biofilm formation in the *hadD*$_{Msm}$KO strain, indicating the key role of *hadD*$_{Msm}$ in this pathway.

Here, we characterized that the contribution of *hadD*$_{Msm}$ to the c-di-GMP-dependent regulation of colony morphology and biofilm formation of *M. smegmatis* and showed that *hadD*$_{Msm}$ is a necessary gene for c-di-GMP-mediated to regulation of the biofilm formation pathway.

## c-di-GMP activates Lsr2$_{Msm}$ to upregulate the expression of *hadD*$_{Msm}$ in *M. smegmatis*

Our previous results showed that *hadD*$_{Msm}$ was the essential gene for c-di-GMP and Lsr2$_{Msm}$ to regulate biofilm formation. Therefore, we hypothesized that c-di-GMP controls biofilm formation by regulating the transcription of *hadD*$_{Msm}$ via Lsr2$_{Msm}$. The impact of c-di-GMP on the expression level of *hadD*$_{Msm}$ was examined through RT–PCR. As shown in Fig. 4D, *hadD*$_{Msm}$ was significantly upregulated in the *ydeH* strain compared to the *ydeH*(mut) strain (Fig. 4D), indicating that high levels of c-di-GMP activated the expression of *hadD*$_{Msm}$. This might be due to the influence of c-di-GMP on the regulation of *hadD*$_{Msm}$ by Lsr2$_{Msm}$. Furthermore, we demonstrated that increasing amounts of c-di-GMP (0.8–16 μM) stimulates the DNA-binding activity of Lsr2$_{Msm}$ through EMSA assays (Fig. S15). To further clarify the regulatory relationship, a chromatin immunoprecipitation (ChIP) assay was conducted to verify whether high levels of c-di-GMP affect the *hadD*$_{Msm}$p-binding activity of Lsr2$_{Msm}$. His-Lsr2$_{Msm}$ and the *ydeH* gene were co-overexpressed in WT *M. smegmatis* to construct a high c-di-GMP-content strain (Msm/*hislsr2*$_{Msm}$-*ydeH*), and *ydeH*(mut) was co-overexpressed to construct a control strain (Msm/*hislsr2*$_{Msm}$-*ydeH*(mut)). As shown in Fig. 4E, the amount of *hadD*$_{Msm}$p precipitated by Lsr2$_{Msm}$ from the *hislsr2*$_{Msm}$-*ydeH*-overexpressing strain was significantly higher than that precipitated from the *hislsr2*$_{Msm}$-*ydeH*(mut)-overexpressing strain. Our results indicated that high levels of c-di-GMP significantly activated the *hadD*$_{Msm}$p-binding activity of Lsr2$_{Msm}$ in *M. smegmatis*.

The effect of intracellular c-di-GMP on the Lsr2$_{Msm}$ regulation of *hadD*$_{Msm}$ was further verified by a β-galactosidase activity assay. We constructed *hadD*$_{Msm}$p-*ydeH*(mut)-*lacZ* or *hadD*$_{Msm}$p-*ydeH*-*lacZ* coexpression plasmids and inserted them into WT and *lsr2*$_{Msm}$KO strains by transformation. Interestingly, the expression of *lacZ* by the *hadD*$_{Msm}$p-*ydeH* overexpression construct in the WT strain was higher than that of the overexpression construct in the *lsr2*$_{Msm}$KO strain. The expression of *lacZ* in the *hadD*$_{Msm}$p-*ydeH*-overexpressing WT strain was significantly upregulated compared with that in the *hadD*$_{Msm}$p-*ydeH*(mut)-overexpressing WT strain (Fig. 4F). These results indicated that c-di-GMP enhanced the expression of *hadD*$_{Msm}$ through Lsr2$_{Msm}$.

In summary, our data suggest that c-di-GMP promotes biofilm formation by activating the *hadD*$_{Msm}$p-binding activity of Lsr2$_{Msm}$ to upregulate the expression of *hadD*$_{Msm}$ in *M. smegmatis*.

## Lsr2 is a conserved c-di-GMP receptor, and the regulation of biofilm formation by *lsr2* and *hadD* is conserved in mycobacteria

Lsr2 is a well-known NAP in *M. tuberculosis* and *M. smegmatis*[16,24,32]. Sequence analysis revealed that Lsr2 is conserved in several important mycobacterial species (*M. tuberculosis*, *M. bovis* BCG, and *M. smegmatis*) (Fig. S16A). The ITC assay showed the specific binding curve of Lsr2$_{Mtb}$ and c-di-GMP, indicating that Lsr2$_{Mtb}$ was also a c-di-GMP signaling receptor in *M. tuberculosis*. The binding stoichiometry between Lsr2$_{Mtb}$ and c-di-GMP was 1:1 (0.923 ± 0.07), and the binding affinity of the interaction (Kd) was $1.074 \times 10^{-7} \pm 0.14$ M (Fig. 5A).

Next, we further showed that *hadD* is highly conserved among important mycobacterial species (Fig. S16B). The HadD protein is

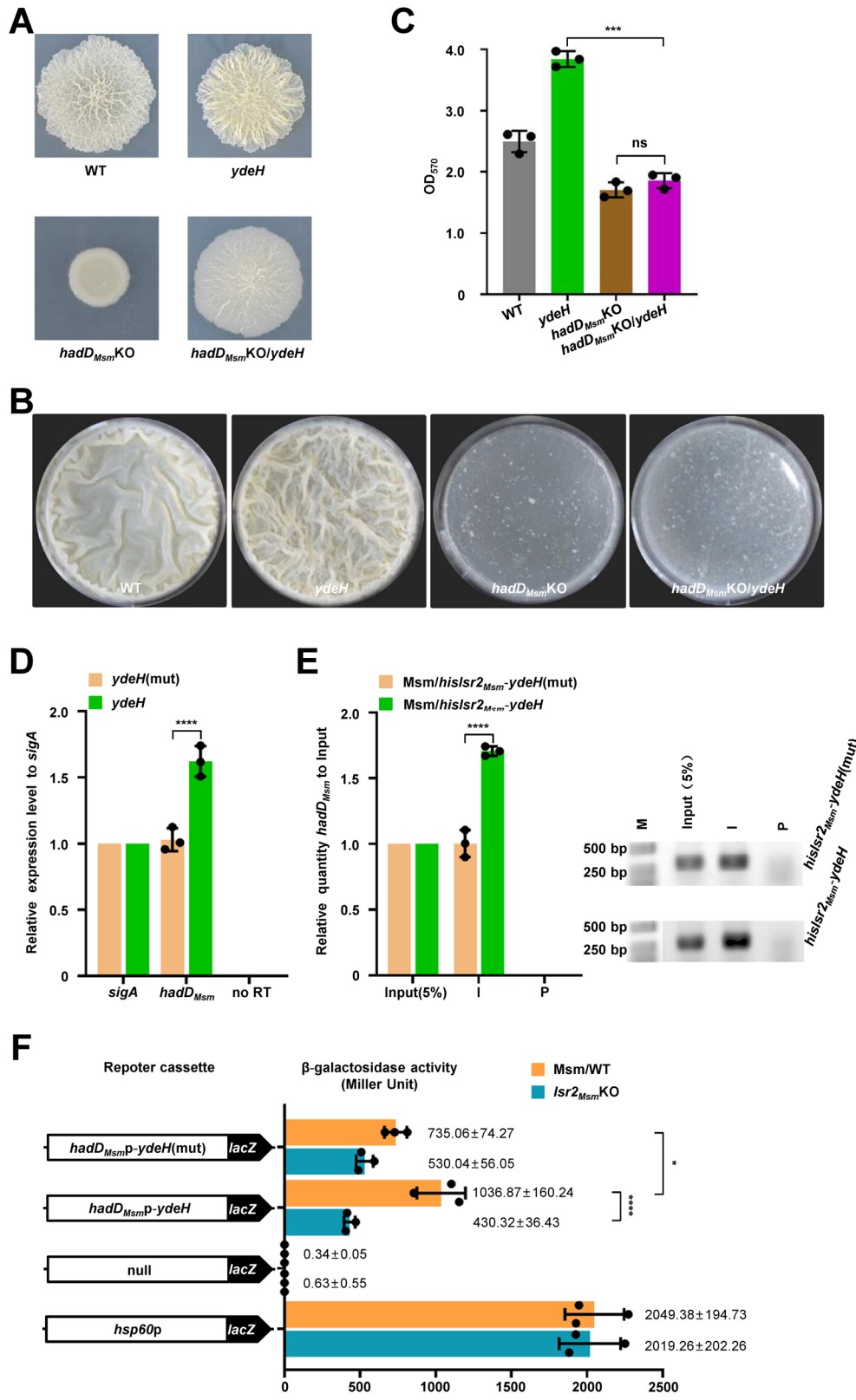

encoded by *rv0504*, and *bcg0547* is entirely identical in *M. tuberculosis* and *M. bovis* BCG, sharing 65.2% identity with *hadD_Msm*. Previous reports and our results confirmed that the regulation of colony morphology and biofilm formation by *hadD_Msm* is conserved[26,27]. The smooth colony phenotype of the *hadD_Msm*KO strain was partially restored to the WT phenotype with the introduction of *hadD_Mtb* of *M. tuberculosis* (*hadD_Msm*KO/*hadD_Mtb*Com) (Figs. 5B, S16C). The results of biofilm culture and quantification are consistent with these findings (Fig. 5C, D). And there was no significant difference in the growth of these strains, indicating that the biofilm formation was not related to the growth (Fig. S1H).

In conclusion, as a receptor for c-di-GMP, NAP Lsr2 is conserved in mycobacteria. In addition, *hadD* from *M. tuberculosis* and *M. smegmatis* relatively conservatively regulates biofilm formation.

**Fig. 4 | c-di-GMP promotes the expression of *hadD*<sub>Msm</sub> by Lsr2<sub>Msm</sub> to affect biofilm formation in *M. smegmatis*.** **A** Spot colony morphology of the wide type and its *ydeH*-overexpressed strains, *hadD*<sub>Msm</sub> knock-out and its *ydeH*-over-expression strains. **B** Biofilm formation of the WT, *ydeH*, *hadD*<sub>Msm</sub>KO, and *hadD*<sub>Msm</sub>KO/*ydeH* strains. **C** Quantitation of biofilm biomass by crystal violet staining of the WT, *ydeH*, *hadD*<sub>Msm</sub>KO, and *hadD*<sub>Msm</sub>KO/*ydeH* strains ($n = 3$, biological replicates). Two-tailed t-tests were performed for statistical analysis (***$p = 0.0004$). **D** RT-PCR quantitation of *hadD*<sub>Msm</sub> transcription in the *ydeH*, *ydeH*(mut)-overexpressed *M. smegmatis* strains. Two-tailed Student's t-tests were performed for statistical analysis (****$p < 0.0001$). RT-PCR and gel analysis were performed under the same conditions and each experiment was performed three independent biological replicates. no RT was genomic DNA contamination control. **E** ChIP assays for the effect of c-di-GMP on the intracellular *hadD*<sub>Msm</sub>p-binding activity of Lsr2<sub>Msm</sub> in *M. smegmatis*. The input (5%) indicated that the supernatant of disrupted cells was diluted to 5%, ChIP using preimmune (P) or immune (I) sera raised against HisLsr2<sub>Msm</sub>. DNA of the input (5%), P, and I were used as temples for PCR (the right panel) ($n = 3$, biological replicates) and RT-qPCR (the light panel). M: DNA Marker. Two-tailed Student's t-tests were performed for statistical analysis (****$p < 0.0001$). **F** β-galactosidase activity assays. The effect of c-di-GMP on Lsr2<sub>Msm</sub> regulates the expression of *hadD*<sub>Msm</sub> was assayed by overexpressing pMV261-*hadD*<sub>Msm</sub>p-*ydeH*-*lacZ* and pMV261-*hadD*<sub>Msm</sub>p-*ydeH*(mut)-*lacZ* plasmids in the Msm/WT, *lsr2*<sub>Msm</sub>KO strains ($n = 3$, biological replicates). None promoter-*lacZ* and *hsp60*p-*lacZ* plasmids overexpression were used as controls. Two-tailed Student's t-tests were performed for statistical analysis (*$p = 0.0414$, ****$p < 0.0001$). Data were presented as mean ± SD of (**C**), (**D**), (**E**), (**F**). The source data were provided in the Source data file.

## Lsr2 regulates biofilm formation by positively regulating the expression of *hadD* in *M. bovis* BCG

Next, we used the vaccine strain *M. bovis* BCG to investigate the effect of Lsr2<sub>BCG</sub> regulation of *hadD*<sub>BCG</sub> on biofilm formation. First, an EMSA demonstrated that Lsr2<sub>BCG</sub> bound well to *hadD*<sub>BCG</sub>p (Figs. 5E and S17). In contrast, Lsr2<sub>Mtb</sub> mutant protein loses the DNA-binding activity (Fig. S18). Then, RT–PCR results showed that the expression of *hadD*<sub>BCG</sub> was obviously reduced in the *lsr2*<sub>BCG</sub>-deficient mutant (*lsr2*<sub>BCG</sub>KO) strain (verified by RT–qPCR, Fig. S19) compared to that in the WT *M. bovis* BCG strain (Fig. 5F). Finally, the smooth colony phenotype of the *lsr2*<sub>BCG</sub>KO strain indicated that Lsr2<sub>BCG</sub> regulated the colony phenotype in *M. bovis* BCG (Fig. 5G).

These data indicate that Lsr2 directly stimulates the expression of *hadD* to control colony morphology and biofilm formation in a conserved manner.

## c-di-GMP promotes positive regulation of *hadD*<sub>BCG</sub> expression by Lsr2

Upon finding the conserved regulation of *hadD*<sub>BCG</sub> by Lsr2<sub>BCG</sub> to control biofilm formation, we further investigated whether the effect of c-di-GMP on this regulation is conserved in the *M. bovis* BCG strain. We significantly increased the intracellular levels of c-di-GMP in *M. bovis* BCG through overexpressing *ydeH* from *E. coli* (Fig. S20)[28]. RT–PCR results showed that the expression level of *hadD*<sub>BCG</sub> significantly increased in the *ydeH*-overexpressing *M. bovis* BCG strain (BCG/*ydeH*) (Fig. 5H). This result indicates that a high level of c-di-GMP triggers the expression of *hadD*<sub>BCG</sub> in *M. bovis* BCG.

ChIP assays were conducted to confirm the regulatory effect of high levels of c-di-GMP on the DNA-binding activity of Lsr2<sub>BCG</sub> in *M. bovis* BCG. We overexpressed *hislsr2*<sub>BCG</sub> and *ydeH* via the pMV261 plasmid to produce a high level of c-di-GMP in the WT *M. bovis* BCG strain (BCG/*hislsr2*<sub>BCG</sub>-*ydeH*) and overexpressed *hislsr2*<sub>BCG</sub> and *ydeH*(mut) to construct a control strain (BCG/*hislsr2*<sub>BCG</sub>-*ydeH*(mut)). As shown in Fig. 5I, Lsr2<sub>BCG</sub> precipitated ~3 times the *hadD*<sub>BCG</sub> promoter from the *hislsr2*<sub>BCG</sub>-*ydeH*-overexpressing strain compared to that from the *hislsr2*<sub>BCG</sub>-*ydeH*(mut)-overexpressing strain. Therefore, this result indicated that high levels of c-di-GMP could significantly promote the *hadD*<sub>BCG</sub>p-binding activity of Lsr2<sub>BCG</sub> in *M. bovis* BCG.

Taken together, these findings suggest that a high level of c-di-GMP activates Lsr2 to positively regulate the expression of *hadD* in mycobacteria. Moreover, the increased synthesis of keto-MA contributing to biofilm formation is conserved in mycobacteria, including the important *M. tuberculosis* and *M. bovis* BCG strains.

## Discussion

Bacteria preferentially transform from suspension culture to a biofilm mode of growth under high levels of c-di-GMP[33]. However, the regulatory pathway of c-di-GMP regulation of biofilm formation in mycobacteria has not been reported. In this study, we report that the robust biofilm formation induced by c-di-GMP depends on the receptor Lsr2, which is a NAP. We successfully demonstrated that Lsr2 positively regulates *hadD* to regulate biofilm formation. Our data revealed that c-di-GMP could bind to Lsr2 at a ratio of 1:1 and enhance the positive regulation of *hadD* expression by Lsr2. HadD is involved in the synthesis of keto-MA and contributes to biofilm formation. We revealed the complete regulatory pathway by which c-di-GMP regulated robust biofilm formation through Lsr2 to affect lipid synthesis. This is also the report of a NAP that plays a role in c-di-GMP functional regulation in mycobacteria.

Since the discovery of the c-di-GMP receptor, multiple regulatory mechanisms of c-di-GMP in diverse bacterial processes have been extensively reported[2,5,34]. In recent years, several c-di-GMP receptors and their physiological regulatory functions in mycobacteria have been confirmed[35]. The c-di-GMP effector, LtmA, stimulates the expression of 37 lipid transport and metabolism genes in *M. smegmatis*[28]. Another c-di-GMP receptor, HpoR, acts as an inhibitor to enhance mycobacterial antioxidant defense[36]. Interestingly, c-di-GMP can integrate LtmA and HpoR to regulate antioxidant processes[36]. In addition, the two-component regulator DevR was characterized as a new c-di-GMP receptor in response to oxidative stress[37]. Here, we found that NAP Lsr2 is a receptor of c-di-GMP that regulates mycobacterial biofilm formation. Lsr2 is a known global transcriptional regulator that contributes to multiple physiological processes[17–19,23,24]. Therefore, our findings indicated that c-di-GMP could combine with NAP Lsr2 to perform wide regulatory functions in mycobacteria.

Biofilms are well known to contribute to the drug resistance of bacteria[38], chronic infections[39,40], and persistent infections[41,42] by pathogenic bacteria. For example, *Pseudomonas aeruginosa* mucoid strains form biofilms in the lungs, leading to cystic fibrosis[43]. *M. tuberculosis* is the causal agent of TB. Chronic TB is difficult to treat with current first-line anti-TB drugs owing to the emergence of multidrug-resistant bacteria. Part of the reason is the formation of biofilms by *M. tuberculosis* in the lung[44,45]. Therefore, target genes involved in biofilm formation are expected to be the focus of TB treatment. In the present study, we identified the regulatory mechanism by which Lsr2 regulates mycobacterial biofilm formation. Lsr2 increases the synthesis of keto-MA in association with biofilm formation by positively regulating the expression of the target gene *hadD*. Importantly, high levels of c-di-GMP trigger the positive regulation of the expression of *hadD* by Lsr2, and the increase in keto-MA synthesis further contributes to robust biofilm formation. Thus, our research indicates that the target gene *hadD* is a key target for drug design for TB prevention and treatment.

Overall, our current data showed that the NAP Lsr2 is a conserved c-di-GMP receptor that regulates biofilm formation in mycobacteria. c-di-GMP positively regulates the expression of *hadD* via Lsr2 to affect biofilm formation, and this pathway is conserved in mycobacteria. Our findings support the model shown in Fig. 6. The increase in the c-di-GMP signal level promotes the positive regulation of *hadD* by Lsr2, leading to an increase in the synthesis of keto-MA. Thereafter, keto-MA

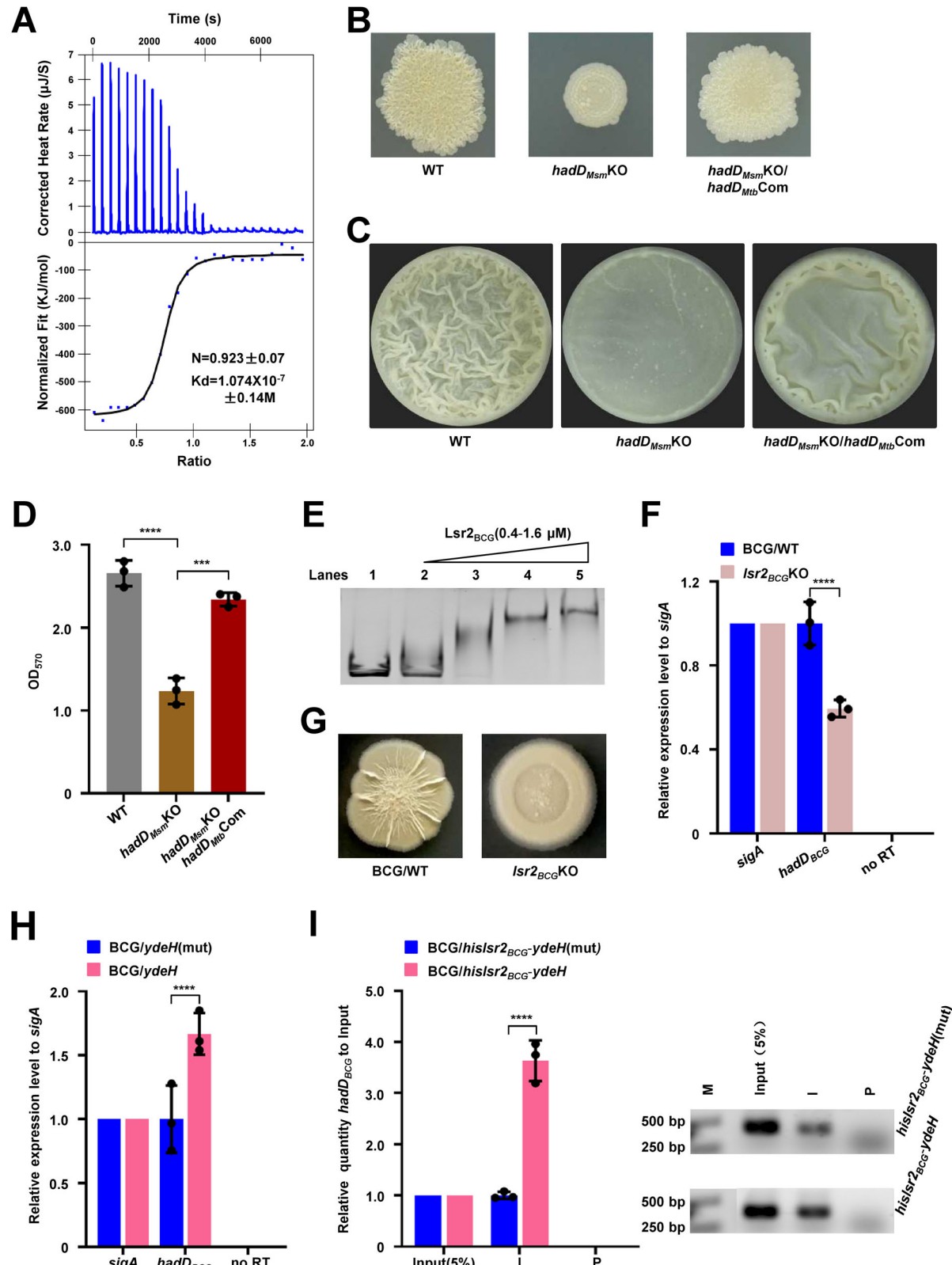

can be converted into trehalose monomycolate (TMM) and trehalose dimycolate (TDM)[46]. TMM, TDM, and MA are important components of the cell wall and affect mycobacterial biofilm formation[26,47]. Our findings extend the regulatory function of c-di-GMP to biofilm formation in mycobacteria. This research also provides a reference for further studies of the extensive signaling pathways of c-di-GMP through NAP.

## Methods

### Expression and purification of recombinant proteins

*lsr2* genes were amplified from genomic DNA of the *M. smegmatis* mc[2] 155 and *M. tuberculosis* strains by polymerase chain reaction (PCR) using appropriate primer pairs (Tsingke Biotech, Beijing). The amplified DNA fragments were cloned into modified pET28a expression vector to obtain recombinant plasmids that were introduced into

**Fig. 5 | The c-di-GMP signal regulatory system is conserved in mycobacteria including *M. tuberculosis*. A** ITC assays for the interaction between Lsr2$_{Mtb}$ and c-di-GMP. Original titration data and integrated heat measurements are shown in the upper and lower plots. **B** Spot colony morphology of the wide type, *hadD*$_{Msm}$ knock-out and *hadD*$_{Mtb}$ complementary strains. **C** Biofilm formation of the WT, *hadD*$_{Msm}$KO, and *hadD*$_{Mtb}$ complementary strains. **D** Quantitation of biofilm biomass of the WT, *hadD*$_{Msm}$KO, and *hadD*$_{Mtb}$ complementary strains by crystal violet staining ($n = 3$, biological replicates). Two-tailed t-tests were performed for statistical analysis (****$p < 0.0001$, ***$p = 0.0004$). **E** EMSA assays for the effect of c-di-GMP on *hadD*$_{BCG}$ promoter DNA-binding activity of Lsr2$_{BCG}$. *hadD*$_{BCG}$p was co-incubated with increasing concentration of Lsr2$_{BCG}$ (lanes 2–5). Three independent experiments were performed. **F** RT-PCR for transcriptional analysis of *hadD*$_{BCG}$ in the WT, *lsr2*$_{BCG}$KO *M. bovis* BCG strains ($n = 3$, biological replicates). Two-tailed Student's t-tests were performed for statistical analysis (****$p < 0.0001$). RT-PCR

and gel analysis were performed under the same conditions. **G** Spot colony morphology of the WT and *lsr2*$_{BCG}$ knock-out BCG strains on 7H10 plates. **H** RT-PCR for transcriptional analysis of *hadD*$_{BCG}$ in the *ydeH*(mut), *ydeH*-overexpressed *M. bovis* BCG strains ($n = 3$, biological replicates). Two-tailed Student's t-tests were performed for statistical analysis (****$p < 0.0001$). **I** ChIP assays for the effect of c-di-GMP on the intracellular DNA-binding activity of Lsr2$_{BCG}$ in the *M. bovis* BCG strains. The input (5%) indicated that the supernatant of disrupted cells was diluted to 5%, ChIP using preimmune (P) or immune (I) sera raised against HisLsr2$_{BCG}$. DNA sample of the input (5%), P, and I were used as temples for PCR (the right panel) ($n = 3$, biological replicates) and were quantified using RT-qPCR (the light panel). M: DNA marker. Two-tailed Student's t-tests were performed to for statistical analysis (****$p < 0.0001$). Data of figures (**D**), (**F**), (**H**), and (**I**) were presented as mean ± SD. The source data were provided in the Source data file.

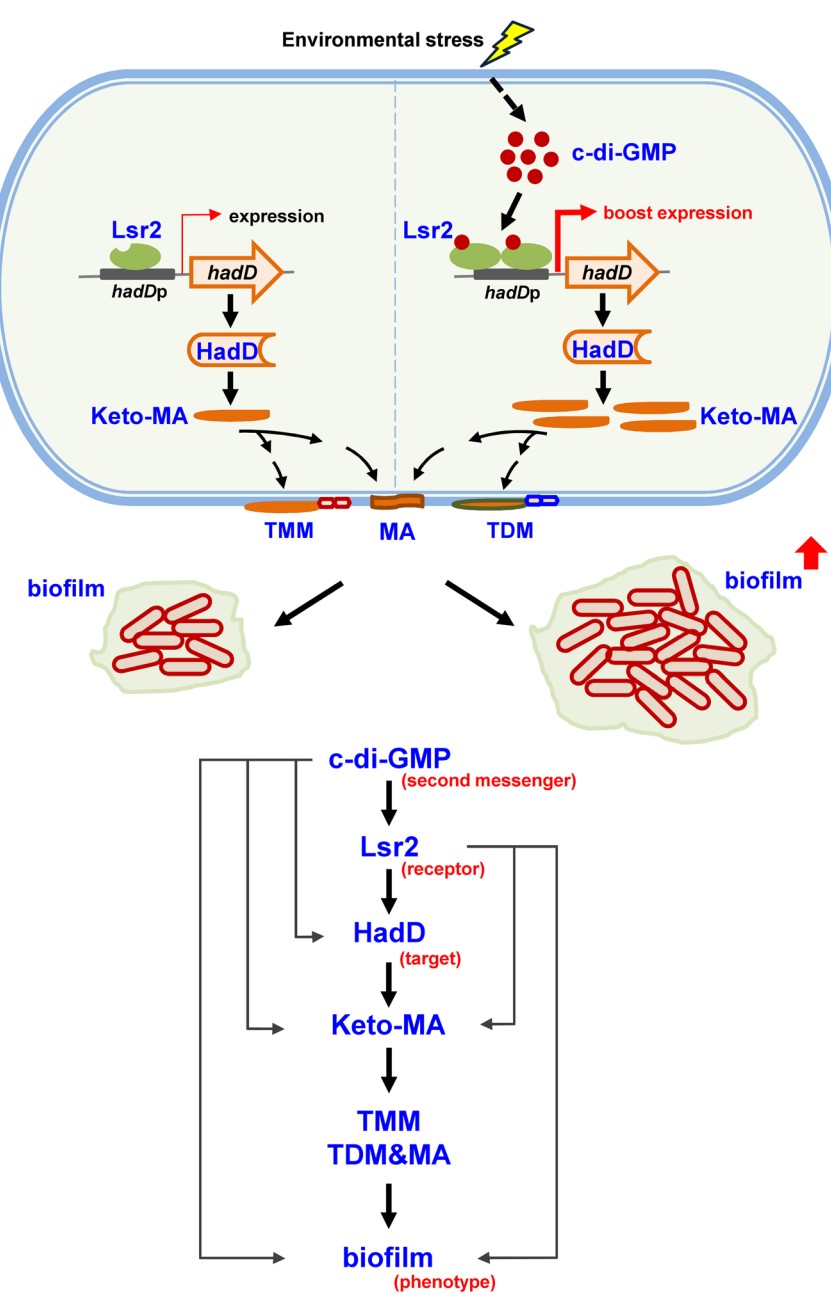

**Fig. 6 |** The model of which c-di-GMP stimulates the expression of *hadD* by Lsr2 to control biofilm formation in mycobacteria.

*E. coli* BL21 (DE3) cells. The cells were cultured to an optical density at 600 nm (OD$_{600}$) of 0.6 in 200 mL LB medium (Trytone 20 g/L (#LP0042B, OXOID), Yeast Extract 10 g/L (#LP0021B, OXOID), NaCl 20 g/L (#A501218, Sangon Biotech)). Protein expression was induced by addition of 0.4 mM isopropyl-β-D-1-thiogalactopyranoside (IPTG) (#I8070, Solarbio) for 12 h at 16 °C. The cells were collected by centrifugation and the protein (His-tag at the C-terminal) was purified by affinity chromatography on Ni-NTA agarose (#SA05101L, Smart-Lifesciences)[28,36]. The column-bound protein was washed with a wash buffer (100 mM Tris–HCl (#T1503, Sigma) pH 8.0, 500 mM NaCl and 40 mM imidazole (#I8090, Solarbio)) and dialyzed using an elution buffer (100 mM Tris–HCl pH 8.0, 500 mM NaCl and 250 mM imidazole). The protein was stored at −80 °C. Purified proteins were verified by sodium dodecyl sulfate-polyacrylamide gel Electrophoresis (SDS-PAGE). Protein concentration was detected by the Bradford method and Nano Drop (Thermo Fisher, USA).

### Generation of a high-level c-di-GMP transposon *M. smegmatis* library and screening the target genes of c-di-GMP

The DGC gene *ydeH* from *Escherichia coli* was cloned and inserted into a modified pMV261 plasmid containing a hygromycin resistance gene to obtain a recombinant plasmid. Subsequently, the recombinant plasmid was introduced into *M. smegmatis* mc²155 by transformation to obtain a high c-di-GMP-content *M. smegmatis* strain (*ydeH*). Then, the transposon system containing a kanamycin resistance gene was introduced into the *ydeH* strain to generate a transposon insertion mutant library of high levels of c-di-GMP of *M. smegmatis* as described previously. The cells of the transposon insertion mutant library were coated on 7H10 agar medium (#262710, BD Difco) supplemented with 0.5% (v/v) glycerol (#A600232, Sangon Biotech), kanamycin (Kan) (#I8020, Solarbio) and hygromycin (Hyg) (#H8080, Solarbio) and cultured at 37 °C for 3–5 days. The strains with altered colony morphology were isolated and cultured in 7H9 medium (#271310, BD Difco) supplemented with 0.2% (v/v) glycerol and 0.05% (v/v) Tween-80 (#A600562, Sangon Biotech) to OD$_{600}$ 1.2. The cells were collected, and genomic DNA was extracted with a bacterial genome extraction kit (#DN1101, Aidlab). Furthermore, the sequences near the insertion site of the transposon were amplified by thermal asymmetric PCR and sequenced to analyze the location of the transposon[19,48]. The primers for *ydeH* overexpression and transposon insertion mutant sequence analysis are shown in the Source data file (Primer list).

### Construction of the deletion mutants and complementation mycobacteria

*lsr2* and *hadD* genes were knocked out by a gene replacement strategy[49]. Upstream and downstream fragments of *lsr2$_{Msm}$*, *hadD$_{Msm}$*, and *lsr2$_{BCG}$* were amplified using corresponding primer pairs. The amplified DNA fragments were cloned into a pMind-derived suicide plasmid that carried a hygromycin resistance gene and a *lacZ* selection market. Subsequently, these recombinant plasmids were inserted into mycobacteria by transformation. *lsr2$_{Msm}$*, *hadD$_{Msm}$*, and *lsr2$_{BCG}$* knockout strains were selected and identified with PCR[50]. The complementation strains were generated using plasmid pMV261. The primers for genes knockout and complementation were shown in the Source data file (Primer list).

### Morphological observation

Mycobacteria for morphological observation was grown to a mid-log phase (OD$_{600}$ 1.0–1.2) in 7H9 medium. The cells were collected by centrifuge and washed with PBS. Next, the cells were re-suspended in 1 mL 25% glutaraldehyde (#G5882, Sigma-Aldrich) and fixed overnight at 4 °C. Then, the cells were freeze-dried after beening dehydrated by ethanol (#A500737, Sangon Biotech). The cells were fixed on the conductive tape and observed by Quattro scanning electron microscopy (OPTON, China)[51]. The cells of crystal violet staining were observed using OLYMPUS CX33 optical microscope (Olympus Corporation, Japan). Mycobacterial colony morphologies were investigated by spot-inoculating 2 μL cultures in 7H10 or LB agar medium at 37 °C for 3–5 days to observe colony morphology. Cultures were diluted to a concentration of 10$^6$ in 100 μL and were plated on 7H10 agar medium to observe single colony morphology.

### Biofilm formation and quantitative analysis

Biofilm formation was performed referring to the previously described procedures[13]. Mycobacteria was cultured to OD$_{600}$ 1.0–1.2 in 7H9 medium and was suspended in a modified M63 medium (addition of 1% (w/v) glucose (#G8150, Solarbio), 0.5% (w/v) casamino acid (#C822594, Macklin), 1 mM MgSO$_4$ (#20025118, Sinopharm) and 0.7 mM CaCl$_2$) (#10005861, Sinopharm)). 5 mL of mycobacterial suspension with an OD$_{600}$ of 0.3 was inoculated into a 12-well PVC microtiter dishes and was incubated at 30 °C without shaking for 5 days to observe biofilms formation at gas–liquid surface. 100 μL mycobacterial suspension with an OD$_{600}$ of 0.1 was used to quantify the biofilm formation by crystal violet staining assay[10,48]. Mycobacterial suspension was incubated for 48 h at 37 °C with shaking at 80 rpm in 96-well PVC microtiter dishes. Biofilms were stained with 1% (w/v) crystal violet (CV) (#C8470, Solarbio) at room temperature for 30 min and were extracted with ethanol (#A500737, Sangon Biotech)/acetone (#W332615, Sigma-Aldrich) (v/v = 80:20). The extracted biofilms were measured at OD$_{570}$ using a TECAN Infinite M200 Pro Nano Quant microplate reader (Mannedorf, Switzerland).

### Screening downstream target genes of c-di-GMP by the CRISPRi platform

CRISPRi platform was constructed through using the plasmid pLJR962[31]. sgRNA scaffolds targeting the *M. smegmatis* genome were designed with two unique *Bsm*B I (#R0739L, NEB) restriction sites. Then, the complementary sgRNA oligos were annealed and ligated into the CRISPRi vector backbone (pLJR962) to construct CRISPRi plasmids. The mixed CRISPRi plasmids of the *M. smegmatis* genome were inserted into the high c-di-GMP-content *M. smegmatis* strain via transformed, and the resulted strains were cultured on 7H10 plates containing 200 ng/mL ATc (#IA5330, Solarbio) for 3–5 days at 37 °C. The empty pLJR962 vector was inserted into the high c-di-GMP-content *M. smegmatis* strain via transformation as a control (pLJR962). The change in the colony phenotype of *M. smegmatis* strain was screened and sequenced with a primer pair to confirm the gene targeting by dCas9-sgRNA complex.

### RNA extraction and RT-PCR assay

Mycobacteria cells were grown to OD$_{600}$ 1.0–1.2 in 7H9 medium and their RNA were extracted by using the RNA extraction kit (#RN0802, Aidlab). 500 ng total RNA was reverse transcribed to cDNA by using a TRUEscript strand cDNA synthesis kit (#PC1803, Aidlab). cDNA was used as a template for RT-PCR to analysis the expression of target gene. RT-PCR assay was performed using a 25 μL conventional PCR reaction and the following protocol: pre-denaturation at 95 °C for 5 min, 40 cycles of 95 °C for 10 s, 60 °C for 30 s, 72 °C for 30 s, and annealing at 72 °C for 5 min. PCR products were analysis by loading on 1.5% agarose (#BY-R0100, Biowest) gel electrophoresis. Images were acquired by a Geldoc scanner (Bio-Rad, USA). The source data were provided in the Source Data file. The primers used in RT-PCR assay were listed in the Source Data file (Primer list).

### Electrophoretic mobility shift assay (EMSA)

The upstream regulatory sequence of *lsr2* or *hadD* clusters used in electrophoretic mobility shift assay (EMSA) were amplified from mycobacterial genome by PCR. Primer pairs for PCR were listed in the Source data file (Primer list). EMSA assays were conducted to detect

the *hadD* promoter-binding activity of Lsr2[50,52] DNA fragments were co-incubated with various concentration of protein diluted in a buffer (50 mM Tris–HCl, pH 7.5, 10% glycerol, 50 mM NaCl) to a total volume of 30 μL at room temperature for 15 min. The mixture was directly loaded on 8% native polyacrylamide gel (#A1020, Solarbio) and separated in a buffer containing 0.5 × Tris-borate-ethylenediaminetetraacetic acid (EDTA) (#E8030, Solarbio) at 150 V for 1 h. Images were acquired by Geldoc scanner (Bio-Rad, USA). c-di-GMP were co-incubated with Lsr2 for 15 min before beening co-incubated with DNA fragments to investigate the effect of c-di-GMP on the interaction between Lsr2 and *hadD* promoter. The source data were provided in the Source data file.

## Chromatin immunoprecipitation assay and real-time qPCR analysis

Chromatin immunoprecipitation (ChIP) assay was performed to investigate the effect of c-di-GMP on the interactions between Lsr2 and *hadD* promoters in vivo[53]. Mycobacteria were grown to $OD_{600}$ 1.0 in 100 mL 7H9 medium, fixed with 1% formaldehyde (#A501912, Sangon Biotech) for 20 min and stopped with 0.125 M glycine (#62011516, Sinopharm) for 5 min. Crosslinked cells were harvested and washed with PBS for three times. The cells were re-suspended in 1 mL Tris-Buffer with Tween-20 and Triton X-100 (20 mM Tris–HCl, 150 mM NaCl, 0.1% Tween 20 (#A600560, BBI), 0.1% Triton X-100 (#A110694, Dimond), pH 7.5). The sample was sonicated on ice and centrifuged. 100 μL of supernatant was saved as input sample. The 900 μL supernatant was incubated with 1:2,000 dilution of mouse 6*His antibodies (#CSB-MA000011M0m, CUSABIO) or preimmune mouse serum (#NS03L, Sigma-Aldrich) for 3 h at 4 °C. Then, the complexes were immunoprecipitated with 20 μL 50% protein A agarose (#17127901, GE) for 1 h. The DNA was dissociated and recovered from the immuno-complex as the sample of ChIP and P[36,54]. The samples of Input, I and P were purified and analyzed by RT-PCR and real-time qPCR using SYBR qPCR Green Master Mix (#PC3301, Aidlab) on QuantStudio 3 Real-Time PCR System (Thermo Fisher, USA). The relative quantity of *hadD*p in ChIP was normalized to the levels of *hadD*p in input. The degrees of change in the levels of *hadD*p were calculated using the $2^{-\Delta\Delta Ct}$ method, and two-tailed Student's t-tests were performed for statistical analysis. The primer pairs used in ChIP assay are listed in the Source data file (Primer list).

## β-galactosidase activity assays

The β-galactosidase activity experiment was performed in the *M. smegmatis* strain by constructing no-promoter/promoter-*lacZ*, and promoter-*yedH/yedH*(mut)-*lacZ* expression plasmids based on pMV261[55]. These plasmids were electroporated into the $lsr2_{Msm}$ knock-out strain and WT *M. smegmatis* strain to obtain recombinant reporter strains. Recombinant strains were grown to $OD_{600}$ 1.0–1.2 in 7H9 medium. The cells were collected and suspended in 600 μL Z buffer (60 mM $Na_2HPO_4 \cdot 12H_2O$ (#A501725, Sangon Biotech), 40 mM $NaH_2PO_4 \cdot 2H_2O$ (#A502805, Sangon Biotech), 10 mM KCl, 1 mM $MgSO_4$ (#A50119, Sangon Biotech), 50 mM β-Mercaptoethanol (#M8210, Solarbio)). 300 μL cell suspension was used to detect $OD_{600}$. Another 300 μL cell suspension was treated with 100 μL chloroform (#13200, Thermo Fisher) and 100 μL 0.1% SDS (#S8010, Solarbio). Then, 200 μL 4 mg/mL substrate 2-nitrophenyl β-D-galactopyranoside (#O8040, Solarbio) was added for β-galactosidase activity detection, and 500 μL $NaCO_3$ (#A500840, Sangon Biotech) was used for stopping experiment. After centrifuging, the absorbance of 200 μL supernatant of reaction was detected at 600 nm, 420 nm, and 550 nm through a TECAN Infinite M200 Pro Nano Quant microplate reader (Mannedorf, Switzerland)[54]. The primers used in β-galactosidase activity experiment are listed in Source data file (Primer list). Two-tailed Student's t-tests were performed for statistical analysis.

## Comparative lipidomic analysis

*M. smegmatis* strains were grown to $OD_{600}$ 1.2 in 7H9 medium and harvested for bacterial lipidomic analysis. Lipid extraction, ultra-performance liquid chromatography, mass spectrometry, and lipid structural analyses were performed at Novo gene (Beijing, China)[48]. Bacterial cells (100 mg) were mixed with 0.75 mL methanol (#R40121, Thermo Fisher), which were vortexed in a glass tube with a Teflon lined cap. 2.5 mL of MTBE (#40477, Thermo Fisher) was added into the mixture and incubated for 1 h at room temperature in a shaker. Next, the upper (organic) phase was collected through centrifugation after adding 0.625 mL $H_2O$. After that, 1 mL of the solvent mixture (MTBE/methanol/water (10:3:2.5, v/v/v)) was added into the lower phase to re-extract. Combined organicphases were dried and dissolved in 100 μL of isopropanol (#022906, Thermo Fisher) for storage. Then, the sample was detected by UHPLC-MS/MS using a Vanquish UHPLC system (Thermo Fisher, USA) coupled with an Orbitrap Q ExactiveTM HF mass spectrometer (Thermo Fisher, USA). Sample was injected into a Thermo Accucore C30 column (150 × 2.1 mm, 2.6 μm) using a 20 min linear gradient at a flow rate of 0.35 mL/min. The mobile phase buffer A was consisted of acetonitrile/water (6:4, v/v) with 10 mM ammonium acetate (#C21999, Thermo Fisher) and 0.1% formic acid (#LS118-4, Thermo Fisher), and buffer B was consisted of acetonitrile (#047138, Thermo Fisher)/isopropanol (#039194, Thermo Fisher) (1:9, v/v) with 10 mM ammonium acetate (#C21999, ThermoFisher) and 0.1% formic acid (#LS118-4, ThermoFisher). The solvent gradient was set as follows: 30% B, initial; 30% B, 2 min; 43% B, 5 min; 55% B, 5.1 min; 70% B, 11 min; 99% B, 16 min; 30% B, 18.1 min. Q ExactiveTM HF mass spectrometer was operated in positive [negative] polarity mode with sheath gas 40 psi, sweep gas 0 L/min, auxiliary gasrate 10 L/min [7 L/min], spray voltage 3.5 kV, capillary temperature 320 °C, heater temperature 350 °C, S-Lens RF level 50, scan range 114–1700 $m/z$, automatic gain control target 3e6, normalized collision energy 22 eV, 24 eV, 28 eV [22 eV, 24 eV, 28 eV], injection time 100 ms, isolation window 1 $m/z$, automatic gaincontrol target (MS2) 2e5, dynamic exclusion 6 s. The Compound Discoverer 3.01 (CD3.1, ThermoFisher) was used to perform peak alignment, peak picking, and quantitation of raw data for each metabolite. The peak intensities were normalized to the total spectral intensity base on these main parameters: retention time tolerance, 0.2 min; actual mass tolerance, 5 ppm; signal intensity tolerance, 30%; signal/noise ratio, 3; and minimum intensity, 100,000. Several databases were used for metabolite identification, including LipidBlast ((https://fiehnlab.ucdavis.edu/projects/LipidBlast), Lipidmaps (http://www.lipidmaps.org) and Mtb LipidDB (https://www.ncbi.nlm.nih.gov/pmc/articles/PMC3073466/). The mass error used was 5 ppm. Each sample was conducted in six independent replicates. Univariate analysis (t-test) was applied to calculate the statistical significance (*p* value) and fold change of the metabolites between the means of the two groups using the statistical software R (version R-3.4.3), Python (version 2.7.6) and CentOS (CentOS release 6.6).

## Detection the levels of c-di-GMP in *M. bovis* BCG

The *yedH*(mut)-overexpression (*yedH*(mut)) and *yedH*-overexpression (*yedH*) of *M. bovis* BCG were constructed for detecting the levels of c-di-GMP. The c-di-GMP of *M. bovis* BCG strains were extracted using a modified protocol according to the following procedures[28]. The strains were cultured to an $OD_{600}$ = 1.2, and cells were collected and washed twice with PBS buffer. Subsequently, cells were suspended in 15 mL $ddH_2O$ and crushed using ultrasound at 450 W for 1 h. The samples were centrifuged. Then, the supernatant was extracted with phenol/chloroform and concentrated to 1.5 mL. Finally, the levels of c-di-GMP were quantified using an ELISA kit (#F20205-B, FANKEWEI).

## Reporting summary

Further information on research design is available in the Nature Portfolio Reporting Summary linked to this article.

## Data availability

The mass spectrometry lipidomic data generated in this study has been deposited in the MetaboLights database (www.ebi.ac.uk/metabolights/MTBLS7053). The processed lipidomic data were available in the Source data file. Mtb LipidDB (https://www.ncbi.nlm.nih.gov/pmc/articles/PMC3073466/) was used for metabolite identification of lipidomic. The absorbance, gels and EMSA data generated in this study have been provided in the Source data file. Source data are provided with this paper.

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

## Acknowledgements

This work was supported by the National Key R&D Program of China 2020YFA0907201 (to Z.G.H. and W.H.L.), Guangxi Science Fund for Distinguished Young Scholars 2022JJG130005 (to W.H.L.), National Natural Science Foundation of China Grants 31870036 (to W.H.L.) and Ba-Gui Scholar Program of Guangxi (to Z.G.H.).

## Author contributions

W.H.L. designed and coordinated this project. X.C.L., X.L., K.W., M.H.G., Y.Z.O., D.T.L., Y.L.X., J.C.Z., L.H.H. and H.Y.Z. conducted experiments. W.H.L. and X.C.L. interpreted results and wrote the manuscript. All authors contributed to the interpretations and conclusions presented.

## Competing interests

The authors declare no competing interests.
