## [Peer Review File · Nature Communications]

Lsr2 acts as a cyclic di-GMP receptor that promotes ketomycolic acid synthesis and biofilm formation in mycobacteriaReviewer #1 (Remarks to the Author):

The authors used a number of genetic and biochemical assays to show interaction of Lsr2 with the second messenger c-di-GMP and how it impacts biofilm production in *M. smegmatis*.

1. Abstract, lines 15-16, seems rather general, do you mean in all bacteria or in mycobacteria? Line 27 correct "messenger's".
2. Main, line 31, I suggest changing "the" for "one" or "a".
3. Main, line 33, I suggest changing "prefer" as it conveys the idea that bacteria sorts of think or decide consciously about this phenomenon.
4. Main, lines 44-45, reference cited (10) does not really support the claim raised by authors here.
5. Main, lines 47-48, it is necessary to clarify that these findings were reported in BCG, not Mtb.
6. Main, lines 50-54, it is necessary to clarify that these findings were reported in *M. smegmatis*, not slow-growing mycobacteria, for the sake of better showing what's known in fast- and slow-growing mycobacteria. The same applies to all the text, i.e., I recommend being explicit as to what mycobacterial species the text discussed is about, including the title.
7. Main, two relevant studies about the participation of Lsr2 in biofilm formation are not considered: PMID: 28628249 and PMID: 32724037.
8. Results, line 105, it is not self-evident how the authors made use of a "transposon insertion mutant library of *M. smegmatis* with high levels of c-di-GMP".
9. Results, lines 101-120: were there other genes, and which ones, identified in their genetic screen? In the absence of testing either the role of DosR/DevR and/or LtmA (PMID: 31270210 and PMID: 23047950) and the known participation of dosR in biofilm production by BCG (PMID 32724037 and PMID: 32238759) the argument of "c-di-GMP relies on the NAP Lsr2Msm to regulate colony phenotype and biofilm formation in *M. smegmatis*" cannot be fully sustained.
10. Results, lines 149, the argument presented in lines 148-149 is an overstatement, as there was still biofilm production in the mutant referred to there.
11. Results, lines 246-247, it is not self-evident that "Previous reports and our results confirmed that the regulation of colony morphology and biofilm formation by hadDMsm is conserved", what reports? What precise evidences? These need to be written and cited accordingly.
12. Results, lines 253-254 "In addition, hadD from *M. tuberculosis* and *M. smegmatis* conservatively regulates biofilm formation." Is an overstatement, as restoring the genes only partially complemented the phenotypes.
13. (a) The claims regarding c-di-GMP binding to Lsr2 and impacting in the phenotypes described must be strengthened by including complemented strains with Lsr2. Further to this, RT-PCR are not the most accurate way of determining changes, moreover, considering the authors were able to conduct high-throughput assays as well as ChIP assays;
- (b) authors did not experimentally confirmed that their so-called high c-di-GMP content strain indeed has increased levels of this second messenger;
- (c) altered colony morphology as starting phenotype to screen for target genes of this second messenger does not necessarily lead to affecting genes where c-di-GMP is required... what if they simple interrupted a gene that was required for optimal growth in the media employed here? This may be entirely c-di-GMP-independent;
- (d) there is no negative control such as a truncated or DNA-binding affected Lsr2 mutant protein;
- (e) there is no inclusion of other controls of transcriptional regulators known to bind c-di-GMP, such as dosR and/or LtmA, what if changes in these genes also lead to the same phenotypes and alter the same pathways found for the Lsr2 mutant?;
- (f) there is no control for EMSA assays or any non-Lsr2 regulated gene nor any change to hadD regulatory sequence that demonstrated that Lsr2 +/- c-di-GMP binding was specific. In fact, EMSA assays seem to have been run in the absence of synthetic c-di-GMP.

Reviewer #2 (Remarks to the Author):

This manuscript deals with a novel receptor of c-di-GMP, Lsr-2, in mycobacteria and attempts to show that the receptor binding to the second messenger c-di-GMP in stoichiometric amount

controls biofilm formation by mycobacteria. Regulation of biofilm formation by a pathogen within human host is a challenging problem both for basic understanding as well designing of drugs to inhibit biofilm growth. In this direction the problem is well chosen. Authors deal with three major candidates for the manipulation of mycobacterial biofilms, Lsr-2, which mediates the level of keto-mycolic acid which in its turn regulates another protein HadD. Overall, authors conclude from their experiments that c-di-GMP and receptor Lsr-2 bind in stoichiometric amount and the binding parameters are linked to lipid synthesis and biofilm formation.

The obvious key experiments will be to determine the stoichiometry between ligand and receptors, manipulate the stoichiometry to observe its effect, and mutate the receptor to control biofilm growth. Both are simple experiments and transposon based mutagenesis studies are attempted to carry out the second part. For the first part, they relied on ITC experiments which yield all thermodynamic parameters. They concluded 1:1 stoichiometry with nano molar binding affinity for ligand and receptor. This values and conclusions are vital and form the basis of the work. However, the curve they report is not a saturation binding, it appears to be allosteric. It requires different mathematical formulation to treat the data.

In addition, there are problem with EMSA experiments (Fig.5E). The figure is unclean and does not appear as saturation binding.

In addition, the language is difficult at places.

Response to the reviewers

Reviewer #1

The authors used a number of genetic and biochemical assays to show interaction of Lsr2 with the second messenger c-di-GMP and how it impacts biofilm production in *M. smegmatis*.

Response: Thanks for your comments. We appreciate it.

1. Abstract, lines 15-16, seems rather general, do you mean in all bacteria or in mycobacteria? Line 27 correct “messeger’s”.

Response: Thanks. It means in bacteria. C-di-GMP is a highly conserved and widely distributed second messenger in bacteria. It has been reported that c-di-GMP regulates many important physiological processes in bacteria, including *Escherichia coli* (PMID: 31836667), *Pseudomonas aeruginosa* (PMID: 23657259), *Acinetobacter baumannii* (PMID: 36191190), and *Shewanella putrefaciens* (PMID: 35315431) etc. The synthesis of lipids is essential for bacterial biofilm formation (PMID: 30782633; PMID: 18179839; PMID: 18160165; PMID: 20154129; PMID: 31551455; PMID: 28198348). Thanks. In the revision, we have corrected “messeger’s” to “messenger’s”.

2. Main, line 31, I suggest changing “the” for “one” or “a”.

Response: We agree. In the revision, we have followed this suggestion to change “the” for “a”.

3. Main, line 33, I suggest changing “prefer” as it conveys the idea that bacteria sorts of think or decide consciously about this phenomenon.

Response: Thanks for the good suggestion. In the revision, we have changed “prefer” for “tend”.

4. Main, lines 44-45, reference cited (10) does not really support the claim raised by authors here.

Response: Sorry. In the revision, we have followed the comment to replace the reference (10) with the references (PMID: 11544235; PMID: 18466296; PMID: 23836904).

5. Main, lines 47-48, it is necessary to clarify that these findings were reported in BCG, not Mtb.

Response: Thanks for the comment. In the revision, we have clarified that these findings were reported in *M. bovis* BCG.

6. Main, lines 50-54, it is necessary to clarify that these findings were reported in *M. smegmatis*, not slow-growing mycobacteria, for the sake of better showing what’s known in fast- and slow-growing mycobacteria. The same applies to all the text, i.e., I recommend being explicit as to what mycobacterial species the text discussed is about,

including the title.

Response: We appreciate your good comments. Indeed, it is necessary to clarify that these findings were reported in fast- or slow-growing mycobacteria for the sake of better showing what's known in fast- and slow-growing mycobacteria. In the revision, we have clarified mycobacterial species in lines 53-57 and all the text.

In our study, we conducted *in vivo* and physiological phenotype experiments in *M. smegmatis* and *M. bovis* BCG. Additionally, we used the homologous gene of *M. tuberculosis* to complement the knock-out strain of *M. smegmatis* for studying the function of homologous gene. Furthermore, the proteins (Lsr2, HadD) are 100% homologous in *M. bovis* BCG and *M. tuberculosis*, and proteins are highly homologous in *M. smegmatis* and *M. tuberculosis* (Fig S5). Hence, if several mycobacterial species appear simultaneously in the text, we used the mycobacteria for statement, including the title.

7. Main, two relevant studies about the participation of *lsr2* in biofilm formation are not considered: PMID: 28628249 and PMID: 32724037.

Response: Thanks for the good suggestion. In the revision, we have added the references (PMID: 28628249 and PMID: 32724037) to the manuscript (line 54).

8. Results, line 105, it is not self-evident how the authors made use of a “transposon insertion mutant library of *M. smegmatis* with high levels of c-di-GMP”.

Response: Sorry. In the revision, we have supplemented the description about the construction and utilization of the “transposon insertion mutant library of *M. smegmatis* with high levels of c-di-GMP” in the Materials and Methods (lines 367-376).

9. Results, lines 101-120: were there other genes, and which ones, identified in their genetic screen? In the absence of testing either the role of DosR/DevR and/or LtmA (PMID: 31270210 and PMID: 23047950) and the known participation of *dosR* in biofilm production by BCG (PMID 32724037 and PMID: 32238759) the argument of “c-di-GMP relies on the NAP *Lsr2*_{Msm} to regulate colony phenotype and biofilm formation in *M. smegmatis*” cannot be fully sustained.

Response: Thanks for the comments. In our genetic screen, we have identified that *PatA* regulates the mycobacterial biofilm formation. This work was published in the journal of *Microbiol. Spectr.* (PMID: 37212713).

In our previous research (PMID:31270210; PMID:23047950; PMID:29982829), we didn't find that *DosR/DevR* and *LtmA* regulate biofilm formation in *M. smegmatis*. We also have detected the effect of *DevR* and *LtmA* on biofilm formation in *M. smegmatis* (the response to reviewer #1-13e, please see the Figures). The results indicate that there are no effect.

It should be noted that c-di-GMP must rely on its downstream different receptors to perform a variety of physiological processes. That is to say, different receptors perform different functions, and each receptor performs the specific function. In our study, the rough colony *ydeH*-overexpressing (high levels of c-di-GMP) strain formed stronger biofilms than smooth colony *lsr2*_{Msm}KO strain and *lsr2*_{Msm}KO/*ydeH* strain (Fig 2A-D).

This indicates that c-di-GMP relies on the NAP Lsr2_{Msm} to regulate colony phenotype and biofilm formation in *M. smegmatis*.

10. Results, lines 149, the argument presented in lines 148-149 is an overstatement, as there was still biofilm production in the mutant referred to there.

Response: Thanks. In the revision, we have corrected the description.

11. Results, lines 246-247, it is not self-evident that “Previous reports and our results confirmed that the regulation of colony morphology and biofilm formation by *hadD*_{Msm} is conserved”, what reports? What precise evidences? These need to be written and cited accordingly.

Response: Thanks for the comments. Previous studies have reported that *hadD* deletion could lead biofilm deficiency in *M. smegmatis* and *M. tuberculosis* (PMID: 29662082; PMID: 32034201). In the revision, we have cited the references (PMID: 29662082; PMID: 32034201) in the revised manuscript (line 256).

12. Results, lines 253-254 “In addition, *hadD* from *M. tuberculosis* and *M. smegmatis* conservatively regulates biofilm formation.” Is an overstatement, as restoring the genes only partially complemented the phenotypes.

Response: Thanks a lot. In the revision, we have corrected the description.

13. (a) The claims regarding c-di-GMP binding to Lsr2 and impacting in the phenotypes described must be strengthened by including complemented strains with *lsr2*. Further to this, RT-PCR are not the most accurate way of determining changes, moreover, considering the authors were able to conduct high-throughput assays as well as ChIP assays;

Response: Thanks for the comments. In the revision, we have followed the suggestions to strengthen the descriptions (lines 112-118, lines 133-136) and experiments about complemented strains with *lsr2* (Fig S2A-C).

In this study, beyond RT-PCR, we also have performed the β -galactosidase activity and ChIP-qPCR experiments to detect the effect of c-di-GMP on the regulation of *hadD* by Lsr2 (Fig 4E-F).

(b) authors did not experimentally confirmed that their so-called high c-di-GMP content strain indeed has increased levels of this second messenger;

Response: We appreciate your comment. In our previous research, it has demonstrated that *ydeH* overexpression could produce high levels of c-di-GMP in *M. smegmatis* (PMID:23047950). Additionally, we have detected the levels of c-di-GMP in *ydeH*-overexpression strains of *M. bovis* BCG (please see the following Figure). The result shows that *ydeH* overexpression also can increase the levels of c-di-GMP in *M. bovis* BCG.

Figure. Detection the levels of c-di-GMP in *M. bovis* BCG. The *ydeH*(mut)-overexpression (*ydeH*(mut)) and *ydeH*-overexpression (*ydeH*) of *M. bovis* BCG were constructed for detecting the levels of c-di-GMP. The c-di-GMP of *M. bovis* BCG strains were extracted using a modified protocol according to the previously procedures (PMID: 23047950). The strains were cultured to an $OD_{600}=1.2$, and cells were collected and washed twice with PBS buffer. Subsequently, cells were suspended in 15 mL ddH₂O and crushed using ultrasound at 450 W for 1h. The samples were centrifuged. Then, the supernatant was extracted with phenol/chloroform and concentrated to 1.5 mL. Finally, the levels of c-di-GMP were quantified using an ELISA kit (FANKEWEI, China). Multiple Student's t-tests were performed for statistical analysis (**P < 0.01).

(c) altered colony morphology as starting phenotype to screen for target genes of this second messenger does not necessarily lead to affecting genes where c-di-GMP is required... what if they simple interrupted a gene that was required for optimal growth in the media employed here? This may be entirely c-di-GMP-independent;

Response: Thanks for the comments. Our results found that high levels of c-di-GMP strain (*ydeH* strain) can form the wrinkled colony compared to *ydeH*(mut) strain (please see the Fig 1 F and G). Hence, to investigate the regulatory mechanism of c-di-GMP on phenotype and screen the target genes, we generated a transposon insertion mutant library of *M. smegmatis* with high levels of c-di-GMP. That is to say, we utilized the altered colony morphology of transposon insertion mutants with high levels of c-di-GMP to screen the target genes which are related to c-di-GMP. The above process is the classic forward genetic screening method for identifying the target genes of c-di-GMP. The detail method was described in the manuscript (lines 367-376). Furthermore, we also constructed a series of recombinant strains about c-di-GMP and screened target gene to verify and detect the phenotype (please see the Fig 2). Ultimately, we determined whether c-di-GMP relies on target genes to regulate the phenotype.

(d) there is no negative control such as a truncated or DNA-binding affected Lsr2 mutant protein;

Response: Thanks a lot. We have followed the suggestion and complemented the negative control for DNA-binding affected Lsr2 mutant protein (please see the following Figures). The previous report showed that the "RGR" amino acids of Lsr2 are responsible for binding DNA (PMID: 20133735). Therefore, we constructed the

Lsr2 mutant protein (RGR to AGA). Our results show that Lsr2 mutant protein loses the DNA-binding activity through performing the EMSA experiment.

Figure. Detection the DNA-binding activity of Lsr2 mutant protein. **A:** Amino acid sequence alignment of Lsr2_{Msm} and Lsr2_{Msm} mutant protein. **B:** EMSA assays for the DNA-binding activity of Lsr2_{Msm} and Lsr2_{Msm} mutant protein. *hadD_{MsmP}* was co-incubated with increasing amounts of Lsr2_{Msm} (lanes 2-5) and Lsr2_{Msm} mutant protein (lanes 7-10). **C:** Amino acid sequence alignment of Lsr2_{Mtb} and Lsr2_{Mtb} mutant protein. **D:** EMSA assays for the DNA-binding activity of Lsr2_{Mtb} and Lsr2_{Mtb} mutant protein. *hadD_{MtbP}* was co-incubated with increasing amounts of Lsr2_{Mtb} (lanes 2-5) and Lsr2_{Mtb} mutant protein (lanes 7-10).

(e) there is no inclusion of other controls of transcriptional regulators known to bind c-di-GMP, such as dosR and/or LtmA, what if changes in these genes also lead to the same phenotypes and alter the same pathways found for the lsr2 mutant?

Response: Thanks for the comments. In our previous studies, we have found that DosR (DevR) and LtmA regulate the antioxidant growth and antibiotic resistance in mycobacteria (PMID: 31270210; PMID: 23047950; PMID: 29982829).

Nevertheless, for addressing your concerns about this issue, we constructed the

recombinant strains using the CRISPRi to down-regulate the expression of *dosR* (*devR*) and *ltmA* in *M. smegmatis*. Then, we have performed phenotype experiments including colony phenotype observation, biofilm formation observation, and biofilm quantification. The results show that *dosR* (*devR*) and *ltmA* can not lead to the same phenotypes compared to *lsr2* in *M. smegmatis* (please see the following Figures).

Figure. The effect of *dosR* (*devR*) and *ltmA* on colony phenotype and biofilm formation in *M. smegmatis*. **A:** Spot colony morphology of the *M. smegmatis* strains on 7H10 medium with 30 mg/mL Kan or 30 mg/mL Kan + 200 ng/mL ATc. The strains include pLJR 962 (containing empty pLJR962 plasmid), *devR*_{Msm} CRISPRi (containing pLJR962-*devR*_{Msm} CRISPRi plasmid) and *ltmA*_{Msm} CRISPRi (containing *ltmA*_{Msm} CRISPRi plasmid). **B:** Biofilm formation observation of the pLJR 962, *devR*_{Msm} CRISPRi, and *ltmA*_{Msm} CRISPRi strains. **C:** Biofilm biomass quantitation of pLJR 962, *devR*_{Msm} CRISPRi, and *ltmA*_{Msm} CRISPRi strains.

(f) there is no control for EMSA assays or any non-Lsr2 regulated gene nor any change to *hadD* regulatory sequence that demonstrated that Lsr2 +/- c-di-GMP binding was specific. In fact, EMSA assays seem to have been run in the absence of synthetic c-di-GMP.

Response: Thanks a lot. Lsr2 is a nucleoid-associated protein (NAP) with non-specific and wide DNA-binding activity, which is the common characteristics of NAP. Hence, in this study, except for EMSA (DNA-binding experiment *in vitro*), we also have performed the β -galactosidase activity and ChIP-qPCR assays *in vivo* to identify the target gene specifically regulated by Lsr2 (please see the Fig 3B, 3C and 4E).

Nevertheless, for addressing your concerns about this issue, we have chosen *ms5038* and *ms4718* (non-Lsr2 regulated genes) and performed EMSA and β -galactosidase activity experiments (please see the following Figure A and B). These results indicate that although Lsr2 can bind the *ms5038p* and *ms4718p* by EMSA assays, it has no regulation for *ms5038* and *ms4718* through β -galactosidase activity experiments. Therefore, the Lsr2 specifically regulates the expression of *hadD*. We have followed the suggestion and performed the EMSA assays in the presence of c-di-GMP (please see the following Figure C). The results show that c-di-GMP promotes the DNA-binding activity of Lsr2.

Figure. The detection of specific regulation of Lsr2 and the effect of c-di-GMP on DNA-binding activity of Lsr2. A: EMSA assays for the DNA-binding activity of Lsr2. The increasing amounts of Lsr2 was co-incubated with *hadD*_{MsmP} (lanes 1-8), *ms5038p* (lanes 9-16) and *ms4718p* (lanes 17-24). B: The β -galactosidase activity experiment to assay for the effect of Lsr2 on the expression of *ms5038* and *ms4718*. None promoter-*lacZ* and *hsp60p-lacZ* were used as controls. The data were presented as Miller units on the right panel. Left column: schematic representation of each plasmid used to construct recombinant strains. Two-tailed Student's t-tests were performed for statistical analysis of three independent biological experiments (ns, no significant difference). C: EMSA assays for the effect of c-di-GMP on DNA-binding activity of Lsr2. *hadDp* was co-incubated with Lsr2 in the absence (lanes 2) or presence of c-di-GMP (0.8-16 μ M) (lanes 3-5).

Reviewer #2

This manuscript deals with a novel receptor of c-di-GMP, Lsr2, in mycobacteria and attempts to show that the receptor binding to the second messenger c-di-GMP in stoichiometric amount controls biofilm formation by mycobacteria. Regulation of biofilm formation by a pathogen within human host is a challenging problem both for basic understanding as well as designing of drugs to inhibit biofilm growth. In this direction the problem is well chosen. Authors deal with three major candidates for the manipulation of mycobacterial biofilms, Lsr2, which mediates the level of keto-mycolic acid which in its turn regulates another protein HadD. Overall, authors conclude from their experiments that c-di-GMP and receptor Lsr2 bind in stoichiometric amount and the binding parameters are linked to lipid synthesis and biofilm formation.

Response: Thanks for your very positive comments. We appreciate it.

The obvious key experiments will be to determine the stoichiometry between ligand and receptors, manipulate the stoichiometry to observe its effect, and mutate the receptor to control biofilm growth. Both are simple experiments and transposon based mutagenesis studies are attempted to carry out the second part. For the first part, they relied on ITC experiments which yield all thermodynamic parameters. They concluded 1:1 stoichiometry with nano molar binding affinity for ligand and receptor. This values and conclusions are vital and form the basis of the work. However, the curve they report is not a saturation binding, it appears to be allosteric. It requires different mathematical formulation to treat the data.

Response: We really appreciate the comments. In our previous reports, we utilized ITC experiments to detect the physical interaction between c-di-GMP and receptor (PMID: 26390966; PMID: 29490073). In this study, we also used ITC experiments to calculate the binding affinity for c-di-GMP and Lsr2. ITC assays were carried out on a Nano ITC Low Volume isothermal calorimeter (TA Instruments, New Castle, DE, USA) according to a previously described procedure (PMID: 26390966). Data were recorded automatically and subsequently analyzed using the NanoAnalyze Software provided by the manufacturer. All the titration curves were fitted to the independent-site binding model. In Fig 2E and 5A, the “Corrected Heat Rate ($\mu\text{J/S}$)” of the curves tend to zero over time and eventually reach equilibrium without any changes. Our results indicate that the binding of c-di-GMP and Lsr2 has reached saturation.

In addition, we also have performed the ChIP-qPCR and β -galactosidase activity assays to verify the functional interaction between c-di-GMP and Lsr2 (please see the Fig 4 E and F). Furthermore, we have complemented the EMSA experiments for detecting the effect of c-di-GMP on DNA-binding activity of Lsr2 (please see the following Figure). The results show that c-di-GMP promotes the DNA-binding activity of Lsr2.

In summary, we have performed the ITC, EMSA, ChIP-qPCR and β -galactosidase activity assays to verify the physical and functional interaction between c-di-GMP and Lsr2.

Figure. The detection for the effect of c-di-GMP on DNA-binding activity of Lsr2. EMSA assays for the effect of c-di-GMP on DNA-binding activity of Lsr2. *hadDp* was co-incubated with Lsr2 in the absence (lanes 2) or presence of c-di-GMP (0.8-16 μM) (lanes 3-5).

In addition, there are problem with EMSA experiments (Fig.5E). The figure is unclean and does not appear as saturation binding.

Response: Thanks for the comments. Since Lsr2 is a nucleoid-associated protein (NAP), the characteristics of its DNA-binding activity are shown in our results. These data are consistent with the previously reported characteristics of Lsr2 binding to DNA (PMID: 20133735; PMID: 33980681). We also have followed the suggestion and tried to increase amounts of Lsr2 to enhance its DNA-binding activity (please see the following Figure). The results show that as the increasing amounts of Lsr2, the shifted DNA formed by the protein-DNA complex are closer to the sample pore. When Lsr2 reaches the high concentration, the shifted DNA of the protein-DNA complex remain in the sample pore. Therefore, the results of the DNA-binding activity of Lsr2 depend on it's characteristics of nucleoid-associated protein (NAP).

Figure. EMSA assays for the DNA-binding activity of Lsr2. **A:** The increasing amounts of *Lsr2_{Msm}* was co-incubated with *hadD_{MsmP}* (lanes 2-10). **B:** The increasing amounts of *Lsr2_{Mtb}* was co-incubated with *hadD_{MtbP}* (lanes 2-10).

In addition, the language is difficult at places.

Response: Thanks a lot. The manuscript has been carefully re-edited by a professional for the English language (English language editing by American Journal Experts).

Reviewer #1 (Remarks to the Author):

I sincerely appreciate your having responded to most of my queries and concerns. I still found two major aspects that I think must be resolved in order for this work to be more suitable of this journal. These are:

1. You have responded to the possible lack of participation of devR/dosR and ltmA in control of biofilm production by mycobacteria. However, how do you reconcile that in PMID: 23047950, the authors found that ltmA regulated colony morphology (where you claim a role for Lsr2 via c-di-GMP binding) and expression of many genes involved in lipid metabolism (which may somewhat also overlap with your own findings)? It is true that in PMID: 23047950 there was no evaluation of biofilm formation, so I think you must also declare this limitation of said reference, which leaves the question open as to whether "c-di-GMP relies on the NAP Lsr2Msm to regulate colony phenotype and biofilm formation in *M. smegmatis*," or that more broadly, c-di-GMP MAY also interact with additional regulators, other than Lsr2, to control biofilm production, given that DosR/DevR has been found to bind this second messenger PMID: 31270210 (no biofilm tested here; however, dosR/devR content was found to be relevant for biofilm formation in *Mtb* (PMID: 32724037)). In short, leaving your work to be entitled as "Lsr2, a novel cyclic di-GMP receptor, links the second messenger to regulate the synthesis of keto-mycolic acid and control biofilm formation in mycobacteria" seems a bit of an overstatement to me, given that: (1) you needed to artificially increase c-di-GMP in your screen (I appreciate your showing this in your rebuttal letter, but, why did you not include this in your revised work?)(2) there is no reference or evidence as to when would *M. smegmatis*, BCG or any other mycobacteria, reach levels of c-di-GMP like the ones reported here, (3) you are not acknowledging the possible participation of other transcriptional regulators already shown to interact with c-di-GMP and regulate biofilm production (DosR) or unknown to do so (ltmA), and (4) coming back to c-di-GMP levels, what are these in biofilm cells produced in your system? I can see that you somewhat experimentally tested this, but I could not find you included these results in your revised manuscript. Given the relevance you are giving to the interaction between Lsr2 and c-di-GMP to control biofilm production, all these results must be shown in an updated manuscript. Furthermore, how do you know that all biofilm-related effects are specific of this type of growth? I think you also need to show what is the growth kinetics, for all strains, in planktonic conditions, to demonstrate whether this is a true effect specific to biofilms or defects are more generally associated to growth defects occurring under any other circumstance.

2. The CRISPRi results partially respond to some of my queries above about dosR/devR and ltmA. However, where is the evidence that your strategy effectively led to lack of transcription/expression of these genes? This must be shown. Also, you must be aware that for lsr2, you had transposon insertion and deletion mutants, which is to be assumed (I could not find RT-PCR or any other test effectively showing lack of lsr2 transcription), whereas as no effect was observed with the CRISPRi strategy for any strain, and no evidence of lsr2 levels was shown, led me to wonder what if the interference did not work at all? In short, lsr2, dosR/devR, and ltmA transcript levels must be shown for all your strains. Furthermore, even though the experiments related to dosR/devR and ltmA were not part of your original study, I think you must include these results for the sake of making a stronger case for Lsr2-c-d-GMP in your biofilm model, which with the evidence providing thus far, does not rule out the possible participation of these (or others) in regulating the same phenotype in mycobacteria.

Reviewer #2 (Remarks to the Author):

The manuscript is revised well and I appreciate the attempts made by the authors to answer all questions raised by the reviewers. I recommend the revised draft for publication.

Response to the reviewers

Reviewer #1

I sincerely appreciate your having responded to most of my queries and concerns. I still found two major aspects that I think must be resolved in order for this work to be more suitable of this journal. These are:

Response: Thanks for your very positive comments. We sincerely appreciate it.

1. You have responded to the possible lack of participation of devR/dosR and ltmA in control of biofilm production by mycobacteria. However, how do you reconcile that in PMID: 23047950, the authors found that LtmA regulated colony morphology (where you claim a role for Lsr2 via c-di-GMP binding) and expression of many genes involved in lipid metabolism (which may somewhat also overlap with your own findings? It is true that in PMID: 23047950 there was no evaluation of biofilm formation, so I think you must also declare this limitation of said reference, which leaves the question open as to whether “c-di-GMP relies on the NAP Lsr2Msm to regulate colony phenotype and biofilm formation in *M. smegmatis*.”, or that more broadly, c-di-GMP may also interact with additional regulators, other than Lsr2, to control biofilm production, given that DosR/DevR has been found to bind this second messenger PMID: 31270210 (no biofilm tested here; however, dosR/devR content was found to be relevant for biofilm formation in Mtb (PMID: 32724037). In short, leaving your work to be entitled as “Lsr2, a novel cyclic di-GMP receptor, links the second messenger to regulate the synthesis of keto-mycolic acid and control biofilm formation in mycobacteria” seems a bit of an overstatement to me, given that: (1) you needed to artificially increase c-di-GMP in your screen (I appreciate your showing this in your rebuttal letter, but, why did you not include this in your revised work?)(2) there is no reference or evidence as to when would *M. smegmatis*, BCG or any other mycobacteria, reach levels of c-di-GMP like the ones reported here, (3) you are not acknowledging the possible participation of other transcriptional regulators already shown to interact with c-di-GMP and regulate biofilm production (DosR) or unknown to do so (LtmA), and (4) coming back to c-di-GMP levels, what are these in biofilm cells produced in your system? I can see that you somewhat experimentally tested this, but I could not find you included these results in your revised manuscript. Given the relevance you are giving to the interaction between Lsr2 and c-di-GMP to control biofilm production, all these results must be shown in an updated manuscript. Furthermore, how do you know that all biofilm-related effects are specific of this type of growth? I think you also need to show what is the growth kinetics, for all strains, in planktonic conditions, to demonstrate whether this is a true effect specific to biofilms or defects are more generally associated to growth defects occurring under any other circumstance.

Response: Thanks for your comments. (1) We constructed the *ltmA*-overexpression strain and found LtmA regulating the colony morphology in *M. smegmatis* (PDIM: 23047950), but deletion *ltmA* had no effect on colony morphology of *M. smegmatis*

(data not shown in PMID: 23047950). In last revision, we detected the effect of LtmA on the biofilm formation of *M. smegmatis* through CRISPRi. The results showed that LtmA didn't affect the biofilm formation in *M. smegmatis*. Furthermore, the target genes regulated by LtmA didn't contain the *hadD* (PMID: 23047950) which was regulated by Lsr2 in this research. Therefore, there is no overlap between the two studies. (2) The DevR was reported as the c-di-GMP receptor only in *M. smegmatis* (PMID: 31270210). All the results were performed in *M. smegmatis* (not in Mtb). It was reported that DevR/DosR affected the biofilm formation in Mtb but not in BCG (PMID: 32724037). In addition, we found that DevR didn't affect the biofilm formation of *M. smegmatis* through CRISPRi in the last revision. (3) In this revision, we have followed your suggestion to modify the title which was entitled as "Lsr2, a novel cyclic di-GMP receptor, regulates the synthesis of keto-mycolic acid and controls biofilm formation in mycobacteria". (4) Sorry, we have followed your suggestion to add the supplemented results to the supplementary materials (Fig. S6D). (5) In last revision, we calculated the relative quantification of the levels of c-di-GMP in the supernatant of the samples. In this revision, we have calculated absolute quantitation, please see the Fig. S6D. (6) We exactly understand what you concern about other transcription factor receptors of c-di-GMP including DevR or LtmA regulating the biofilm formation. Our previous research reported that DevR and LtmA regulate the antioxidant growth and antibiotic resistance in *M. smegmatis* (PMID: 23047950; PMID: 31270210), respectively. And it was reported that DevR/DosR affected the biofilm formation only in Mtb but not in BCG (PMID: 32724037) or *M. smegmatis*. In addition, we found that DevR and LtmA didn't affect the biofilm formation of *M. smegmatis* through constructing the recombinant strains using the CRISPRi in the last revision. In summary, the c-di-GMP receptors DevR and LtmA didn't regulate the biofilm formation in *M. smegmatis*. That is to say, c-di-GMP regulates biofilm formation independent of DevR and LtmA in *M. smegmatis*. Nevertheless, in this study, the rough colony *ydeH*-overexpressing (high levels of c-di-GMP) strain formed stronger biofilms than smooth colony *lsr2_{Msm}*KO strain and *lsr2_{Msm}*KO/*ydeH* strain (Fig. 2A-D). The phenotype of *lsr2_{Msm}*KO/*ydeH* is more similar to that of *lsr2_{Msm}*KO. This is strong evidence and indicates that c-di-GMP largely relies on the NAP Lsr2_{Msm} to regulate colony phenotype and biofilm formation in *M. smegmatis*. (7) We have added the complementary results to this revised manuscript (Fig. S2; Fig. S4; Fig. S5B; Fig. S6D). (8) Thanks a lot. For addressing your concerns about this issue, we detected the growth kinetics for all strains. The results show that the biofilms formation are not related to growth (please see the following figures).

Figure. Growth kinetics for all strains. The detection of growth curves were performed according to the previous procedures with some modifications (PDIM: 23047950). Cultures were obtained and plated on 7H10 medium to determine colony-forming units at the indicated times. **A:** Growth kinetics of the WT, *ydeH* and *ydeH*(mut) strains. **B:** Growth kinetics of the WT, *Isr2_{Msm}* knock-out and *Isr2_{Msm}* complementary strains. **C:** Growth kinetics of the *ydeH*, *Isr2_{Msm}* knock-out and *Isr2_{Msm}*KO/*ydeH* strains. **D:** Growth kinetics of the WT, *hadD_{Msm}* knock-out and *hadD_{Msm}* complementary strains. **E:** Growth kinetics of the WT, *Isr2_{Msm}* knock-out, *hadD_{Msm}* knock-out and its *Isr2_{Msm}* overexpression strains. **F:** Growth kinetics of the WT, *ydeH*, *hadD_{Msm}* knock-out and its *ydeH* overexpression strains. **G:** Growth kinetics of the WT, *hadD_{Msm}* knock-out and its *hadD_{Mtb}* complementary strains. **H:** Growth kinetics of the Msm/pLJR 962, *devR_{Msm}* CRISPRi and *ltmA_{Msm}* CRISPRi strains

2. The CRISPRi results partially respond to some of my queries above about *dosR/devR* and *ltmA*. However, where is the evidence that your strategy effectively

led to lack of transcription/expression of these genes? This must be shown. Also, you must be aware that for *lsr2*, you had transposon insertion and deletion mutants, which is to be assumed (I could not find RT-PCR or any other test effectively showing lack of *lsr2* transcription), whereas as no effect was observed with the CRISPRi strategy for any strain, and no evidence of *lsr2* levels was shown, led me to wonder what if the interference did not work at all? In short, *lsr2*, *dosR/devR*, and *ltmA* transcript levels must be shown for all your strains. Furthermore, even though the experiments related to *dosR/devR* and *ltmA* were not part of your original study, I think you must include these results for the sake of making a stronger case for Lsr2-c-d-GMP in your biofilm model, which with the evidence providing thus far, does not rule out the possible participation of these (or others) in regulating the same phenotype in mycobacteria.

Response: Thanks for your comments. For addressing your concerns about this issue, we detected the transcription levels of *lsr2*, *devR*, and *ltmA* through RT-qPCR in relative recombinant strains. The results show that the transcription levels of *lsr2*, *devR*, and *ltmA* are all decreased in recombinant strains, respectively (please see the following figures).

Figure. Detection of expression levels of target genes in gene knock-out and CRISPRi strains. RT-qPCR assays for determining the relative expression levels of *lsr2_{Msm}*/*lsr2_{BCG}*, *devR_{Msm}* and *ltmA_{Msm}*. Expression levels of genes were normalized using the *sigA* gene as an invariant transcript. Two-tailed Student's t-tests were performed for statistical analysis of three independent biological experiments (**** P < 0.0001). no RT was genomic DNA contamination control. Data were analyzed using the $2^{\Delta\Delta Ct}$ method. **A:** Relative expression levels of *lsr2_{Msm}* in the Msm/WT and *lsr2_{Msm}* KO strains. **B:** Relative expression levels of *lsr2_{BCG}* in the BCG/WT and *lsr2_{BCG}* KO strains. **C:** Relative expression levels of *devR_{Msm}* and *ltmA_{Msm}* in the Msm/pLJR 962, *devR_{Msm}* CRISPRi or *ltmA_{Msm}* CRISPRi strains.

Reviewer #2

The manuscript is revised well and I appreciate the attempts made by the authors to answer all questions raised by the reviewers. I recommend the revised draft for publication.

Response: Thank you very much. We appreciate it.